# Simplified Synthesis of Renieramycin T Derivatives to Target Cancer Stem Cells via β-Catenin Proteasomal Degradation in Human Lung Cancer

**DOI:** 10.3390/md21120627

**Published:** 2023-11-30

**Authors:** Zin Zin Ei, Satapat Racha, Masashi Yokoya, Daiki Hotta, Hongbin Zou, Pithi Chanvorachote

**Affiliations:** 1Department of Pharmacology and Physiology, Faculty of Pharmaceutical Sciences, Chulalongkorn University, Bangkok 10330, Thailand; hushushin@gmail.com (Z.Z.E.); satapatto@gmail.com (S.R.); 2Center of Excellence in Cancer Cell and Molecular Biology, Faculty of Pharmaceutical Sciences, Chulalongkorn University, Bangkok 10330, Thailand; 3Interdisciplinary Program in Pharmacology, Graduate School, Chulalongkorn University, Bangkok 10330, Thailand; 4Department of Pharmaceutical Chemistry, Meiji Pharmaceutical University, 2-522-1, Noshio, Kiyose, Tokyo 204-8588, Japan; yokoya@my-pharm.ac.jp (M.Y.); 0eh3264t133443f@au.com (D.H.); 5College of Pharmaceutical Sciences, Zhejiang University, Hangzhou 310058, China; zouhb@zju.edu.cn

**Keywords:** stem cells, β-catenin, lung cancer, Renieramycin T, DH_32

## Abstract

Cancer stem cells (CSCs) found within cancer tissue play a pivotal role in its resistance to therapy and its potential to metastasize, contributing to elevated mortality rates among patients. Significant strides in understanding the molecular foundations of CSCs have led to preclinical investigations and clinical trials focused on CSC regulator β-catenin signaling targeted interventions in malignancies. As part of the ongoing advancements in marine-organism-derived compound development, it was observed that among the six analogs of Renieramycin T (RT), a potential lead alkaloid from the blue sponge *Xestospongia* sp., the compound DH_32, displayed the most robust anti-cancer activity in lung cancer A549, H23, and H292 cells. In various lung cancer cell lines, DH_32 exhibited the highest efficacy, with IC_50_ values of 4.06 ± 0.24 μM, 2.07 ± 0.11 μM, and 1.46 ± 0.06 μM in A549, H23, and H292 cells, respectively. In contrast, parental RT compounds had IC_50_ values of 5.76 ± 0.23 μM, 2.93 ± 0.07 μM, and 1.52 ± 0.05 μM in the same order. Furthermore, at a dosage of 25 nM, DH_32 showed a stronger ability to inhibit colony formation compared to the lead compound, RT. DH_32 was capable of inducing apoptosis in lung cancer cells, as demonstrated by increased PARP cleavage and reduced levels of the proapoptotic protein Bcl2. Our discovery confirms that DH_32 treatment of lung cancer cells led to a reduced level of CD133, which is associated with the suppression of stem-cell-related transcription factors like OCT4. Moreover, DH_32 significantly suppressed the ability of tumor spheroids to form compared to the original RT compound. Additionally, DH_32 inhibited CSCs by promoting the degradation of β-catenin through ubiquitin–proteasomal pathways. In computational molecular docking, a high-affinity interaction was observed between DH_32 (grid score = −35.559 kcal/mol) and β-catenin, indicating a stronger binding interaction compared to the reference compound R9Q (grid score = −29.044 kcal/mol). In summary, DH_32, a newly developed derivative of the right-half analog of RT, effectively inhibited the initiation of lung cancer spheroids and the self-renewal of lung cancer cells through the upstream process of β-catenin ubiquitin–proteasomal degradation.

## 1. Introduction

Lung cancer is recognized as the most aggressive and prevalent cause of cancer-related mortality [1]. Despite advancements in different cancer treatment protocols, the survival rates of patients diagnosed with lung cancer in all stages have not seen improvement [2]. The decreased survival rate among lung cancer patients is attributed to cancer relapse and the development of drug resistance in cancer treatment [3,4]. Targeted therapy, involving precise medications directed at molecular targets, has been proposed to enhance treatment responses in lung cancer patients [5,6,7].

Research has demonstrated the existence of a distinct group of cancer cells known as CSCs [8,9]. These CSCs have characteristics associated with stemness, such as the capacity for self-renewal and differentiation. As a result, they contribute to tumorigenicity, metastasis, drug resistance, and cancer recurrence [10,11]. While advancements have been made in developing novel strategies for anti-cancer immunotherapy, the current limitations persist, including low treatment response rates and treatment-related side effects. Moreover, experimental evidence currently supports the effectiveness of CSC-targeted therapy in impeding metastasis and reducing the probability of cancer recurrence in patients [12]. 

CSCs can be identified by several means, including the detection of CD133 (Prominin-1). The increased expression of CD133 in lung cancer cells is associated with resistance to chemotherapy and unfavorable clinical prognosis [13]. In addition, the stem cell phenotype can be manifested through transcription factors that regulate stemness, such as OCT4 (octamer-binding transcription factor 4), NANOG (Nanog homeobox), and SOX2 ((sex determining region Y)-box 2) [14]. Likewise, CD44, a transmembrane glycoprotein, plays a crucial role in the functions of CSCs, including self-renewal, resistance to apoptosis, migration, cell differentiation, and proliferation within the microenvironment of cancer cells. As per the existing literature, CD44 functions as the marker initiating tumors in lung cancer cells, as demonstrated in both in vitro and in vivo tests [15]. 

The β-catenin signaling pathway is a conserved signaling axis that plays a role in various physiological processes, including proliferation, differentiation, apoptosis, migration, invasion, and resistance to therapy [16]. In lung cancer, it was shown that β-catenin and its related signaling mediate the increase in transcription factors associated with the CSC phenotype [17,18]. Within the cytoplasm, the aggregation of β-catenin is rendered inactive by polymerized disheveled (Dvl) in the β-catenin destruction complex. In the degradation complex, the phosphorylation of β-catenin is facilitated by glycogen synthase kinase 3β (GSK3β) and casein kinase 1α (CK1α), leading to its ubiquitination and subsequent degradation through the proteasome [16]. As β-catenin signaling is critical for proliferation and CSC control in cancer [19], this has led to belief in the success and prosperity of drug development focused on β-catenin-targeted therapies in treating cancer [20]. 

RT, a tetrahydroisoquinoline alkaloid isolated from the blue sponge *Xestospongia* sp., [21] has been identified as cytotoxic to various cancer cell lines, including human colorectal carcinoma cell line HCT116 [22], lung cancer QC56 [23], human pancreas adenocarcinoma ascites metastasis AsPC1, and breast cancer cell line T47D [21].

Even though RT demonstrates anti-cancer activity, its intricate and large structure poses a considerable challenge for synthesis on a large scale. To overcome this challenge, scientists synthesized RT as the lead compound, creating right-half and left-half derivatives of RT [24]. These derivatives have demonstrated a reduction in the synthesis steps required for the right-half parent compound, featuring a straightforward structure that is easy to synthesize [25]. Hence, it is necessary to investigate the signaling pathway through which these RT derivatives exert targeted anti-cancer activity. Consequently, our primary research emphasis lies in exploring the inhibitory effects of RT derivatives (DH_32) on CSCs within human lung cancer tissues through the induction of β-catenin depletion via ubiquitin–proteasomal degradation. Our discovery could contribute to the potential use of RT derivatives (DH_32) as an innovative β-catenin-targeted agent for the treatment of CSC-driven cancers. 

## 2. Results

### 2.1. Synthesis of Right-Half RT Derivative DH_32

The various previously synthesized *N*-alkylated compounds, **3,** with phenol in the E ring were oxidized to the corresponding *p*-quinone, **4,** by using salcomine. In this study, we also synthesized **4d** with an alkyne instead of an aromatic ring and **4f** with a phthalimide, with the substituent at the C-1 position of phthalascidine. Synthesis of **4f** was performed through amino alcohol **2e**. The lactam carbonyl was partially reduced to generate aminal, which was then treated with KCN and acetic acid to give the aminonitrile **2e** as a single diastereomer. Next, **2e** was debenzylated by reaction with BCl_3_ in the presence of pentamethylbenzene, and the corresponding phenol **3e** was synthesized [26]. Oxidation of **3e** with O_2_ in the presence of salcomine afforded p-quinone **4e** (Figure 1A). Finally, phthalimide moiety was introduced by utilizing the Mitsunobu procedure [27]. The structures of all synthesized compounds (Figure 1B) were analyzed and confirmed by detailed spectral analysis, including 2D-NMR.

### 2.2. Assessing Cytotoxicity through an MTT Assay When Screening for Right-Half RT Analogs

An MTT assay was employed to assess the cytotoxicity characteristics of the right-half analog of the RT derivative in comparison to the parental RT compound. The cytotoxicity of the right-half derivative of the RT analog was examined by subjecting NSCLCs (A549, H23, and H292) to varying concentrations for a 24 h treatment. Afterward, the determination of cell viability was conducted. The results suggested that, when compared to the parent RT compound, the right-half RT derivatives, particularly DH_32, exhibited a greater potent cytotoxic effect on these lung cancer cell lines. In lung cancer cell lines A549, H23, and H292, DH_32 demonstrated the greatest efficacy among the RT derivatives, with IC_50_ values of 4.06 ± 0.24, 2.07 ± 0.11 μM, and 1.46 ± 0.06 μM, respectively. In comparison, the IC_50_ values for RT were 5.76 ± 0.23 μM, 2.93 ± 0.07 μM, and 1.52 ± 0.05 μM in A549, H23, and H292, respectively. The IC_50_ values for DH_18, DH_21, DH_35, DH_38, and DH_39 ranged from 28 to 77 μM, all of which were higher than those of the parent RT compound (Figure 2A).

Anti-cancer drugs can induce cell death through a process known as apoptosis. The condensed nucleus and fragmented chromatin observed through Hoechst 33342 staining serve as indicators of the early phases of apoptosis. Propidium iodide (PI) was utilized as a costain with Hoechst 3342, revealing red fluorescence to denote dead cells. After a 24 h treatment with 0.5 μM of the right-half RT analogs (DH_18, DH_21, DH_32, DH_35, DH_38, and DH_39), the presence of condensed and fragmented nuclei was assessed in lung cancer cells. Among these, the treatment of lung cancer cells with DH_32 significantly elevated the number of apoptotic cells compared to the parent RT compound. The results revealed that DH_32 induced a greater percentage of apoptosis in the A549, H23, and H292 cell lines (29.33 ± 0.51%, 39.29 ± 1.88%, and 34.13 ± 0.97%, respectively) compared to the RT compound (24.95 ± 1.32%, 33.41 ± 0.53%, and 29.63 ± 0.47%, respectively) (Figure 2B). 

Furthermore, lung cancer cells treated with DH_32 exhibited elevated PI staining, indicating an increase in dead cells compared to parent RT compounds. The percentage of dead cells in A549, H23, and H292 cells treated with DH_32 was higher (23.38 ± 0.96%, 35.07 ± 1.67%, and 29.82 ± 1.26%, respectively) compared to the parent RT compound, which showed lower values (21.60 ± 0.99%, 31.15 ± 0.48%, and 26.31 ± 1.04%, respectively) (Figure 2B). Hence, DH_32 was chosen for additional experiments aimed at identifying the targeted signaling pathway for its anti-cancer activity.

To assess the selectivity of DH_32 for lung cancer cells, experiments were conducted wherein human dermal papilla cells (DP) and a non-tumorigenic epithelial cell line derived from human bronchial epithelium cells (BEAS2B) were treated with DH_32. The IC_50_ values for DP and BEAS2B following DH_32 treatment was 34.17 ± 1.22 μM and 5.06 ± 0.22 μM, respectively, as illustrated in Figure 2C. The IC_50_ values for DP and BEAS2B after treatment with the parent compound, RT, were 13.69 ± 0.33 μM and 1.18 ± 0.02 μM, respectively. These findings indicate that DH_32 exhibits lower toxicity in normal cells compared to RT. 

Furthermore, the selectivity index (SI) values of DP cells for tumor cells when treated with DH_32 were 8.42, 16.51, and 23.40 for A549, H23, and H292, respectively. The BEAS2B cell selectivity index values for tumor cells with DH_32 was 1.25, 2.44, and 3.47 for A549, H23, and H292, respectively. A favorable selectivity index (SI) greater than 1.0 suggests that a compound is more effective against lung cancer cells than normal cells. 

### 2.3. DH_32 Inhibits Proliferation, Decreases Colony Formation, and Affects Apoptosis-Related Proteins

The antiproliferative effect of DH_32 was explored at low concentrations ranging from 0 to 100 nM. The lung cancer cells were cultured in a growth medium with or without DH_32 for 72 h, and cell viability was assessed every 24 h. The proliferation assay indicated that A549 cells treated with DH_32 exhibited a decreased proliferation rate, starting at 24 h with doses of 50 and 100 nM, compared to RT. In contrast, RT treatment did not affect the proliferation of A549 cells at this time and dose. At 48 h, A549 cells treated with DH_32 showed a significant reduction in cell proliferation at doses of 25, 50, and 100 nM. However, cells treated with RT did not exhibit any effect at doses of 25 and 50 nM. In the case of other cells, specifically H23 and H292 cells, treatment with DH_32 led to a substantial reduction in the proliferation rate beginning at 24 h with a 25 nM dose. In contrast, cells treated with RT did not exhibit any change in their proliferation rate at 24 h with a 25 nM dose. At the subsequent time points of 48 h and 72 h, DH_32- and RT-treated cells showed consistent and comparable reductions in the proliferation rates of the cells (Figure 3A). Hence, the proliferation rate of lung cancer cells treated with DH_32 markedly diminished in a dose-dependent manner, particularly at the concentration of 25 nM, starting from 24 h.

The colony formation assay serves as a fundamental method for assessing an individual cell’s ability to form colonies, specifically focusing on growth cancer cells. Following a 24 h treatment with DH_32 and RT (at concentrations of 0, 25, 50, and 100 nM) on NSCLC cells (A549, H23, and H292), the drugs were subsequently removed, and the cells were permitted to undergo colony growth for an additional 7 days. The colony formation, as indicated by crystal violet staining, exhibited a markedly higher inhibition rate at 90.83 ± 0.1% for A549 cells treated with 25 nM of DH_32 compared to 72.58 ± 0.07% for RT (25 nM). The cells treated with H23 and H292 showed a greater suppression in the percentage of colony formation at 78.49 ± 0.08% and 79.74 ± 0.14%, respectively, when exposed to a 25 nM dose of DH_32 compared to RT, for which the inhibition rates were 58.75 ± 0.01% and 32.22 ± 0.05%, respectively. At a 25 nM dosage, DH_32 significantly suppressed colony formation across all cell lines compared to RT, as illustrated in Figure 3B.

To verify the apoptosis mechanism, Western blot analysis was utilized to explore levels of apoptotic-related proteins and apoptosis markers, including cleaved PARP, Bcl-2, and Bax. The findings indicated a six-fold increase in cleaved PARP in response to DH_32 (50 nM) compared to the RT treatment in A549 cells. The cleaved PARP levels showed a substantial increase in response to 50 nM of DH_32 in H23 and H292 cells. The levels of anti-apoptotic Bcl-2 protein exhibited a significant reduction in a dose-dependent manner following treatment with DH_32 in A549, H23, and H292 cells. The levels of the pro-apoptotic protein Bax showed a two-fold increase in A549 cells treated with DH_32 compared to the parent compound. However, the levels of this pro-apoptotic protein remained unaltered in H23 and H292 cells treated with DH_32 (Figure 3C).

### 2.4. Inhibitory Effect of DH_32 on CSCs in Various Lung Cancer Cell Lines

CSCs present promising drug targets, offering a potentially effective approach to treating cancer [12]. To test whether DH_32 potentially suppresses lung CSCs, we first generated a CSC-rich population from lung cancer cell lines as described in the Section 4. The immunofluorescence method was employed to determine the CD133 levels in 3D spheroid cells enriched with CSCs. The 3D CSC spheroids were treated with DH_32 and the control non-treated spheroids and the DH_32-treated spheroids were subjected to immunofluorescent analysis for CSC determination. 

The CSC spheroids of A549, H23, and H292 cells were treated with a low concentration of DH_32 (50 nM), and the expression levels of CSC markers and stem cell transcription factors (CD133, CD44, and OCT4) were analyzed by the immunofluorescence method. Figure 4A,B show that DH_32 treatment in CSCs caused a dramatic decrease in the levels of CD133, CD44, and OCT4, implying the depletion of stem cell phenotypes. For comparison, RT was used as a positive control, and results show that RT caused lesser effects on the CSC population. As it was previously demonstrated that the CSC population can resist conventional chemotherapy, including cisplatin [28,29], we similarly treated the spheroids with cisplatin and measured the CD133, CD44, and OCT4 levels. Treatment with cisplatin (50 nM) could not alter CSC markers in these spheroids (Figure 4A).

### 2.5. DH_32 Suppression of CSC-like Phenotypes in Lung Cancer Cells

To further confirm the CSC suppression, we tested whether DH_32 could decrease the expression of stem-cell-related transcription factors. The self-renewal characteristic of cancer cells can be attributed to the phenotypes of stem cells, characterized by increased expression levels of stemness transcription factors such as OCT4 (octamer-binding transcription factor 4), NANOG (Nanog homeobox), and SOX2 (sex-determining region Y-box 2) [14]. 

The mRNA expression levels of stemness transcription factors were assessed using the real-time RT-qPCR method. Our results indicated that the mRNA expression levels of OCT4, NANOG, and SOX2 in A549 cells treated with DH_32 showed a notable decrease to 0.5-fold, 0.4-fold, and 0.3-fold, respectively, at a dosage of 50 nM (Figure 5A). Similarly, the mRNA results indicated that DH_32 significantly decreased the transcription factors of OCT4, NANOG, and SOX2 in both H23 and H292 cells (Figure 5A). 

Next, the protein expression of such transcription factors was determined. A549, H23, and H292 cells were treated with low concentrations of DH_32 (25, 50, and 100 nM) for 24 h, and then the expression levels of the stem cell markers (CD133, CD44, and ALDH1A1), as well as the stem cell transcription factors (OCT-4, NANOG, and SOX2), were analyzed by immunofluorescence assays. The heatmap illustration in Figure 5B indicates that DH_32 treatments caused a significant reduction in stem cell markers and transcription factors in A549, H23 and H292 cells compared to cancer cells treated with the RT parent compound (Figure 5B). Therefore, treatment with the RT analog DH_32 serves as a stem-cell-targeted therapy for human lung cancer cells. 

The A549, H23, and H292 cells were treated with low concentrations of DH_32 (25, 50, and 100 nM) for 24 h, and the expression levels of the CD133, CD44, OCT4, and ALDH1A1 proteins were analyzed by western blot analysis. Figure 5C shows that DH_32 caused a dramatic decrease in the CSC markers CD133, CD44, OCT4, and ALDH1A1 in all A549, H23, and H292 cells (Figure 5C). These findings suggest that DH_32 may have the ability to selectively target stem cells in lung cancer.

### 2.6. DH_32 Destabilizes β-Catenin and Facilitates Proteasomal Degradation

β-Catenin is known to regulate upstream signaling for the expression of cancer stem cell phenotypes in lung cancer cells [19]. To investigate the mechanism of action of DH_32, A549, H292, and H23 cells were similarly treated with DH_32 (0–100 nM) for 24 h, and the expression levels of β-catenin were determined by immunofluorescence. The results showed that DH_32-treated cancer cells exhibited a decrease in the fluorescence signal of β-catenin. However, RT compounds at a concentration of 25 nM exhibited an unaltered level of β-catenin in the cell (Figure 6A). Next, the level of β-catenin proteins was confirmed by Western blot analysis. Figure 6B shows that DH_32 treatments caused a significant decrease in the β-catenin protein level in A549, H23, and H292 cells. 

It is well known that the function of β-catenin is tightly controlled by the processes of ubiquitin–proteasomal degradation [30]. The degradation level of β-catenin was assessed in lung cancer cells treated with DH_32 by tagging ubiquitin using immunoprecipitation analysis. We performed an immunoprecipitation assay for the β-catenin–ubiquitin complex and found that treatment of the A549, H23, and H292 cells with DH_32 (50 nM) significantly increased the amount of β-catenin–ubiquitin complex. In A549 and H292 cells treated with DH_32, an increase of 2.5-fold and 3.5-fold in the level of β-catenin–ubiquitin complex, respectively, was observed compared to cells treated with RT. However, in H23 cells treated with DH_32, there was a two-fold increase in the formation of the β-catenin–ubiquitin complex compared to cells treated with RT. Hence, the findings indicate that the mechanism of action of the right-half RT analog, DH_32, involves inducing β-catenin degradation through ubiquitin–proteasomal degradation, thereby reducing the formation of cancer stem cells (Figure 6C). 

### 2.7. Analysis of Compound DH_32 Interactions with the GSK-3β Protein

To further uncover the possible mechanism by which DH_32 regulates β-catenin stability, we utilized molecular docking analysis to investigate the possible interaction of the compound with β-catenin regulatory protein GSK-3β. PF-04802367, a selective binding of the GSK-3β protein, was employed as a reference compound in the docking method (PDB: 5K5N). Redocking of the native ligand (PF-04802367) at the ATP site of GSK-3β was performed, and the reported RMSD between the native ligand and docked conformation was 1.8431 Å, which is considered a successful docking protocol (Figure 7A). Based on the molecular docking study, the grid score of DH_32 (−38.421 kcal/mol) was higher than that of PF-04802367 (−49.199 kcal/mol), suggesting a lower binding affinity to GSK-3β (Table 1). Furthermore, the effective potency of GSK-3β is reliant on a crucial interaction involving a hydrogen bond with Val135. This interaction is recognized as pivotal for achieving optimal GSK-3β activity [31]. The result revealed that PF-04802367 exhibited interaction with Val135 in the hinge region, while DH_32 does not create a hydrogen bond (Figure 7B,C). Compared with the footprint analysis results for DH_32 and PF-04802367 regarding specific VDW or ES energies, we find that DH_32 makes less favorable VDW energies compared to PF-04802367 at specific residues such as Phe67, Lys85, Val110, Leu130, Leu132, Tyr134, Val135, Pro136, Glu137, Arg141, and Cys199. A slightly unfavorable ES energy with Lys85, Leu132, Tyr134, Val135, Pro136, and Asn186 was also uncovered (Figure 7D). Our findings demonstrated that DH_32 does not bind to GSK-3β in a significant way.

### 2.8. Binding Interaction of DH_32 with β-Catenin

R9Q, a direct inhibitor of β-catenin, was utilized as a reference compound in the docking method (PDB: 7AFW). The redocking of the native ligand (R9Q) at the binding site of β-catenin was performed, resulting in a reported RMSD of 0.4852 Å, which falls within an acceptable range (<2 Å) (Figure 8A). According to the molecular docking study, the grid score of DH_32 (−35.559 kcal/mol) was lower than that of R9Q (−29.044 kcal/mol), indicating a higher binding affinity to β-catenin than the reference compound (Table 2). DH_32 adopted a similar binding mode as R9Q, with an interface between armadillo repeats 2 (residues 205–210) and 3 (residues 243–251) (Figure 8B,C). The footprint analysis confirmed that DH_32 interacts favorably with Asn204, Ala211, Lys242, Met243, Ser246, and Val251 (Figure 8D). This suggests that the contribution of van der Waals (VDW) energy is the most significant factor in the interaction between the ligands and residues at the β-catenin binding site. This finding strongly supports the notion that DH_32 significantly inhibits β-catenin via direct action.

## 3. Discussion

Renieramycins are tetrahydroisoquinoline alkaloids that have been isolated from several marine organisms, including the Thai blue sponge *Xestospongia* [32,33], as well as from *Reniera* [34], *Cribrochalina* [35], and *Neopetrosia* [36]. Numerous scientific studies have demonstrated the anti-cancer properties of renieramycins. 

RT exhibited apoptotic effects in lung cancer cells by specifically targeting the anti-apoptotic protein Mcl-1 and promoting the degradation of Mcl-1 through proteasomes [37]. A derivative of RT, known as 5-O-acetyl-renieramycin T, may exhibit increased sensitivity to cisplatin-induced apoptosis in lung cancer cells [38]. The modification of RT to 5-O-(N-Boc-L-alanine) enables it to bind to Akt, leading to its inhibition and a reduction in the suppression of CSCs [39].

While RT exhibits significant anti-cancer potential, its primary drawback lies in the complex synthesis of the RT compound and its modified structures, particularly when attempting to achieve large-scale production due to the complexity of the RT structure and the numerous synthesis steps involved. For this reason, we endeavor to eliminate specific functional groups from the RT structure that do not contribute to its powerful biological effects.

According to the existing literature, all renieramycins display robust anti-cancer properties. However, in our study, we have explored various derivatives of RT. RT has previously been synthesized by modifying it into two components, specifically the right-half and left-half of RT. Previous research indicated that derivatives of the right-half of RT exhibited higher anti-cancer activity compared to those derived from the left-half in different cancer cell lines, including colorectal cancer (HCT116) and lung cancer cells (QG56), as determined by comparing IC_50_ values in cytotoxicity tests [24,25].

The nitrile functional group plays a significant role in the anti-cancer activity of RT’s structure. Therefore, we synthesized and evaluated the activity of CDE ring model compounds corresponding to the right half of RT that exhibit potent anti-cancer activity [25]. As a result, it was confirmed that strong anti-cancer activity was shown when the p-quinone ring, which is the same as in RT, was used as the aromatic ring corresponding to the E ring of RT [21]. Modifications to the structure of the RT compound at Ring E involved the substitution of various groups, namely pyridine 2-methyl, pyridine 3-methyl, naphthalen-2-yl methyl, prop-2yn-1-yl, 2-hydroxy ethyl, and 1,3-dioxoisoindolin-2-yl ethyl, resulting in the creation of different RT analogs known as DH_18, DH_21, DH_32, DH_35, DH_38, and DH_39, respectively (Figure 1B).

All the mentioned RT analogs were used to treat NSCLC cells to assess their IC_50_ values and calculate the percentages of apoptosis. The IC_50_ value for DH_18 was approximately 30 μM, while DH_21, DH_35, and DH_38 exhibited similar IC_50_ values of around 50 μM. Among the compounds, DH_39 displayed the highest IC_50_ value and resulted in a low percentage of apoptosis in lung cancer cells. Of these compounds, DH_32, an RT analog with naphthalen-2-yl methyl substitution, exhibited a lower IC_50_ value and a higher percentage of apoptosis at a dose of 0.5 μM compared to the original RT compound in NSCLCs (Figure 2A,B). The IC_50_ values for DH_32 treatment in NSCLCs (A549, H23, and H292) were 4.06 ±0.24 μM, 2.07 ± 0.11 μM, and 1.46 ± 0.06 μM, respectively, in comparison to the IC_50_ values of the parent RT compound.

To investigate the impact of DH_32 on cell proliferation, low concentrations of DH_32 (0–100 nM) were administered to NSCLCs. The proliferation rate was evaluated at 0, 24 h, 48 h, and 72 h using an MTT assay. At a dose of 25 nM, DH_32 exhibited a significant decrease in cell proliferation rate compared to the RT parent compound in H23 and H292 cells at 24 h. Furthermore, when A549 cells were treated with DH_32, the proliferation rate decreased at doses of 25 nM and 50 nM, compared to RT treatment at 48 h (Figure 3A).

The colony formation assay indicated that cancer cells treated with DH_32 exhibited a dose-dependent reduction in the number of colonies formed compared to cancer cells treated with RT (Figure 3B). Both the proliferation assay and colony formation assay revealed that DH_32 compounds were more efficient in inhibiting the proliferation rate of cancer cells than the original RT compound, particularly at lower concentrations, especially at a dosage of 25 nM.

We confirmed the presence of apoptosis in cancer cells by assessing apoptosis-related protein markers, including PARP, the anti-apoptotic protein Bcl-2, and the pro-apoptotic protein Bax, using Western blot analysis. The findings indicated that the increase in the levels of PARP cleavage serves as an indicator of apoptosis induction in lung cancer cells treated with DH_32. Additionally, the treatment of DH_32 to lung cancer cells resulted in a dose-dependent decrease in the expression level of the anti-apoptotic protein Bcl-2. Furthermore, the expression level of the pro-apoptotic protein Bax significantly increased when lung cancer cells were treated with 50 nM of DH_32, as shown in Figure 3C.

Numerous treatment approaches have been explored for NSCLCs, with a particular emphasis on targeting CSCs and modulating self-renewal pathways as a key strategy in the treatment of lung cancer [40]. Extensive evidence suggests that the presence of self-renewal capacity and the overexpression of stem cell transcription factors have been associated with accelerated cancer cell growth, increased invasion, metastasis, drug resistance, and unfavorable clinical outcomes [41,42,43]. Three-dimensional spheroids, which mimic in vivo conditions and replicate tumor characteristics, represent a more robust and superior model for in vitro drug screening, facilitating the discovery of new anti-cancer compounds [44]. Culturing cells in ultra-low attachment plates, which promote cell detachment, has been observed to stimulate and preserve the self-renewal ability of lung cancer cell populations rich in CSCs [45,46]. Furthermore, the expression of stem cell markers (CD133, CD44, and ALDH1A1) and stem cell transcription factors (OCT4, NANOG, and SOX2) is notably elevated in the formation of spheroids, which contributes to the self-renewal and proliferation of these spheroids [47]. In our experiments, it was evident that the size of spheroid formations decreased when treated with DH_32 (50 nM). Additionally, there was a notable reduction in the levels of not only stem cell markers CD133 (Figure 4A) and CD44 (Figure 4B), but also the stem cell transcription factor OCT4, (Figure 4B) compared to spheroids treated with RT (50 nM) in lung cancer. Furthermore, spheroids treated with DH_32 exhibited lower expression levels of stem cell markers compared to spheroids treated with cisplatin (50 nM), which served as the positive control (Figure 4A–C). Therefore, the DH_32 compound is promisingly targeted for the treatment of CSC-rich sub-populations in lung cancer. 

The anti-cancer efficacy of DH_32 was demonstrated through its ability to inhibit CSCs derived from different human lung cancer cell lines, including A549, H23, and H292 cells. A549 and H23 cells are derived from lung adenocarcinoma. This is the primary subtype of NSCLCs, accounting for 80% of lung cancers [48]. A549 human lung cancer cells, characterized by KRas and CDKN2A mutations, and H23 cells exhibit mutations in p53 and KRas. H292 cells are obtained from metastatic lung mucoepidermoid carcinoma, a cancer that originates in the salivary glands lining the tracheobronchial tree of the lung and is characterized by CDKN2A mutations [49,50].

The mutations in the KRas and CDKN2A oncogenes, as well as the p53 tumor suppressor gene, result in the continuous proliferation of cancer cells and contribute to the formation of cancer stem cells [51,52,53,54,55,56]. Hence, it is essential to validate the findings from these selected cell line studies in relevant in vitro models to discover new insights into targeted populations rich in CSCs in NSCLCs. 

The induction of stem cell transcription factors OCT4 and NANOG enhances CSCs’ properties and elevates the malignancy of lung adenocarcinoma [57]. Furthermore, the SOX2 transcription factor plays a role in both tumor development and the maintenance of pluripotency in human lung cancer [58]. In our research, lung cancer cells treated with DH_32 exhibited a reduction in the mRNA expression levels of the OCT4, NANOG, and SOX2 transcription factors. This led to an inhibitory effect on CSC-rich spheroids, decreasing their self-renewal capacity and tumor formation in lung cancer stem cells (Figure 5A). We evaluated the influence of RT analogs, specifically DH_32, on the protein levels associated with stem cell phenotypes in human lung cancer cells using both immunofluorescence and Western blot analysis. The fluorescence signals of stem cell markers, including CD133, CD44, and ALDH1A1, were notably reduced by approximately 0.5-fold, and the stem cell transcription factors OCT4, NANOG, and SOX2 also decreased by around 0.5-fold in A549, H23, and H292 cells when treated with DH_32, particularly at a dose of 50 nM (Figure 5B). Furthermore, the protein expression levels of stem cell markers such as CD133, CD44, and ALDH1A1 demonstrated a significant decrease in a dose-dependent manner in A549, H23, and H292 cells treated with DH_32 (Figure 5C).

The activation of β-catenin is directly associated with increased tumor metastasis, migration, invasion, higher disease prevalence, malignant progression, poor prognosis, and elevated mortality rates in lung cancer patients [16]. The dysregulated activation of the β-catenin pathway results in the nuclear accumulation of β-catenin and the transcription of other oncogenes, including c-Myc and Cyclin D-1, which play a role in the initiation and advancement of cancer cell progression [59]. Hence, the β-catenin signaling pathway is a direct target for enhancing the efficacy of lung cancer treatment.

Based on the immunofluorescence analysis, it was observed that DH_32 treatment in human lung cancer cells led to a decrease in the expression of β-catenin levels compared to the parent compound RT. Furthermore, the treatment of lung cancer cells with DH_32 showed a noticeable degradation of β-catenin in cell membranes when compared to untreated cells (Figure 6A). Additionally, the expression level of the β-catenin protein was diminished in lung cancer cells treated with DH_32 (Figure 6B). Hence, we monitored the levels of the β-catenin–ubiquitin complex and observed a significant increase in the formation of the degradation complex in lung cancer cells treated with DH_32 (Figure 6C). The findings indicate that DH_32 primarily targets the degradation of β-catenin and holds promise as a drug for the treatment of NSCLCs.

Hence, this research indicates that the reduction in stem cell markers such as CD133, CD44, and ALDH1A1, as well as stem cell transcription factors OCT4, NANOG, and SOX2, by DH_32 is likely achieved through the proteasomal degradation of β-catenin (Figure 9). Based on our experiments and molecular docking analysis, it was revealed that the RT analog DH_32 directly interacts with target β-catenin in lung cancer cells (Figure 8). Therefore, the inhibitory impacts on the CSC characteristics and stem cell transcription factors achieved by DH_32 are mediated through the proteasomal degradation of β-catenin.

Hence, the effectiveness of DH_32 at a low concentration (25 nM) on tumor initiation and cancer metastasis should be evaluated in in vivo models to validate its potential for clinical investigation. Furthermore, DH_32 serves as the primary compound for targeted anti-cancer strategies and exhibits greater potency than the parent RT compound, particularly at very low doses.

## 4. Materials and Methods

### 4.1. Cell Culture

Cell lines of human NSCLC cells, including A549, H23, and H292, were procured from the American Type Culture Collection (ATCC, Manassas, VA, USA). The ATCC accession numbers for A549, H23, and H292 cells (Homo sapiens—Human) are CVCL_0023, CVCL_1547, and CVCL_0455, respectively. A549 cells were grown in Dulbecco’s Modified Eagle’s Medium (DMEM) (Gibco, Grand Island, NY, USA), while H23 and H292 cells were cultured in Rosewell Park Memorial Institute (RPMI) 1640 medium (Gibco, Grand Island, NY, USA). The non-tumorigenic epithelial cell line from human bronchial epithelium cells (BEAS2B) (Homo sapiens—Human) is CVCL_0168. Human dermal papilla cells (DP) were obtained from Applied Biological Materials Inc. (Richmond, BC, Canada). Both BEAS2B and DP cells were cultured in Dulbecco’s Modified Eagle’s Medium (DMEM) (Gibco, Grand Island, NY, USA). Both media were supplemented with 10% fetal bovine serum (FBS), 2 mM L-glutamine (Gibco, Gaithersburg, MA, USA), and 100 units/mL of antibiotic–antimycotic. The cells were maintained in a 37 °C incubator with 5% CO_2_. The cells reached 80% confluence before proceeding with further experiments in this study.

### 4.2. Reagents and Antibodies

3-(4,5-Dimethylthiazol-2-yl)-2,5-diphenyltetrazolium bromide (MTT), Hoechst 33342, propidium iodide (PI), dimethyl sulfoxide (DMSO), and crystal violet were acquired from Sigma-Aldrich, Co. (St. Louis, MO, USA). Bovine serum albumin (BSA), Alexa Fluor™ 594 goat anti-rabbit IgG (H+L) conjugated secondary antibody, and Alexa Fluor™ 488 goat anti-mouse IgG (H+L) conjugated secondary antibody were obtained from Invitrogen by Thermo Fisher Scientific (Carlsbad, CA, USA).

All primers for OCT4, NANOG, SOX2, and GAPDH were sourced from Eurofins Genomics (Plantside Drive Louisville, KY, USA). Primary rabbit monoclonal antibodies against OCT4 (cat no: ab19857), NANOG (cat no: ab80892), SOX2 (cat no: ab97959), and CD133 (cat no: ab19898) were acquired from Abcam (Waltham, MA, USA). Rabbit monoclonal antibodies for PARP (cat no: 9532), Bcl-2 (cat no: 4223), Bax (cat no: 5023), ALDH1A1 (cat no: 36671), β-catenin (cat no: 8480), and β-actin (cat no: 4970), as well as the mouse monoclonal antibody for CD44 (cat no: 3570), were sourced from Cell Signaling (Beverly, MA, USA). The rabbit secondary antibodies anti-rabbit (cat no: 7074) and anti-mouse IgG (cat no: 7076) were provided by Cell Signaling (Beverly, MA, USA). 

### 4.3. Synthesis of Right-Half RT Analogs

To a solution of phenol, **3a** (9.6 mg, 26.3 µmol), in THF (0.7 mL) was added salcomine (9.5 mg, 26.3 µmol, 1.0 eq.) at room temperature, and the reaction mixture was stirred for 3 h under an O_2_ atmosphere (Figure 1). The reaction mixture was filtered through a cellulose pad and washed with EtOAc. The filtrate was concentrated in vacuo to give a residue. The residue was purified by SiO_2_ flash column chromatography (CH_2_Cl_2_: MeOH = 99:1) to afford compound 4a (8.0 mg, 80%) as a yellow amorphous substance. [α]^27^_D_ −12.8 (c 0.16, CHCl_3_); IR (KBr) 3012, 2943, 28493, 1653, 1615, 1434, 1374, 1308, 1236, 1155, 1005, 951, 864, 755, 667 cm^−1^; ^1^H-NMR (400 MHz, CDCl_3_) δ 8.51 (1H, dt, J = 4.8, 1.2 Hz, 4′-H), 7.58 (1H, td, J = 7.7, 1.2 Hz, 6′-H), 7.16 (1H, dd, J = 7.7, 4.8 Hz, 5′-H), 7.12 (1H, d, J = 7.7 Hz, 7′-H), 4.00 (3H, s, 9-OCH_3_), 3.87 (1H, s, 1-H), 3.77 (2H, s, 1′-H), 3.67 (1H, d, J = 2.1 Hz, 4-H), 3.31 (1H, dt, J = 7.4, 2.1 Hz, 5-H), 3.03 (1H, dd, J = 11.5, 3.3 Hz, 2-H), 2.72 (1H, dd, J = 20.8, 7.4 Hz, 6-H), 2.61 (1H, d, J = 11.5 Hz, 2-H), 2.33 (3H, s, 11N-CH_3_), 2.17 (1H, d, J = 20.8 Hz, 6-H), 1.99 (3H, s, 8-CH_3_); ^13^C-NMR (100 MHz, CDCl_3_) δ: 186.9 (C), 182.3 (C), 156.5 (C), 155.3 (C), 149.7 (CH), 141.0 (C), 137.2 (C), 136.6 (CH), 128.8 (C), 122.7 (CH), 122.7 (CH), 115.9 (C), 61.0 (CH_3_), 60.6 (CH_2_), 58.4 (CH), 54.6 (CH), 51.8 (CH_2_), 51.3 (CH), 41.5 (CH_3_), 20.8 (CH_2_), 8.8 (CH_3_); EIMS *m*/*z* (%) 378 (M+, 6), 220 (23), 219 (100), 218 (36), 204 (24), 176 (11), 160 (38), 93 (20); HRMS (EI) *m*/*z* 378.1689 (M+, calcd for C_21_H_22_N_4_O_3_, 378.1692). The ^1^H-NMR and ^13^C-NMR figures are shown in Appendix A.

To a solution of phenol, **3b** (10.9 mg, 29.9 µmol), in THF (0.8 mL) was added salcomine (11.0 mg, 29.9 µmol, 1.0 eq.) at room temperature, and the reaction mixture was stirred for 2 h under an O_2_ atmosphere (Figure 2). The reaction mixture was filtered through a cellulose pad and washed with EtOAc. The filtrate was concentrated in vacuo to give a residue. The residue was purified by SiO_2_ flash column chromatography (CH_2_Cl_2_: MeOH: Et_3_N = 98:1:1) to afford compound 4b (2.3 mg, 20%) as a yellow amorphous substance. [α]^26^_D_ +20.1 (c 0.09, CHCl_3_); IR (KBr) 2929, 1652, 1307, 1236, 1157, 757, 490, 476, 458, 448, 442, 435, 423, 409 cm^−1^; ^1^H-NMR (400 MHz, CDCl3) δ 8.51 (1H, dd, J = 4.8, 1.1 Hz, 5′-H), 8.43 (1H, d, J = 1.6 Hz, 3′-H), 7.47–7.45 (1H, m, 7′-H), 7.21 (1H, dd, J = 7.7, 4.8 Hz, 6′-H), 4.00 (3H, s, 9-OCH_3_), 3.87 (1H, s, 1-H), 3.66 (1H, d, J = 13.6 Hz, 1′-H), 3.60 (1H, d, J = 13.6 Hz, 1′-H), 3.55 (1H, d, J = 1.8 Hz, 4-H), 3.31 (1H, d, J = 7.2 Hz, 5-H), 2.96 (1H, dd, J = 11.5, 3.1 Hz, 2-H), 2.72 (1H, dd, J = 20.6, 7.2 Hz, 6-H), 2.58 (1H, d, J = 11.5 Hz, 2-H), 2.33 (3H, s, 11N-CH_3_), 2.11 (1H, d, J = 20.6 Hz, 6-H), 2.00 (3H, s, 8-CH_3_); ^13^C-NMR (100 MHz, CDCl_3_) δ: 186.7 (C), 182.2 (C), 155.3 (C), 150.1 (CH), 149.5 (CH), 140.9 (C), 137.1 (C), 136.2 (CH), 131.7 (C), 128.8 (C), 123.6 (CH), 115.4 (C), 61.0 (CH_3_), 58.0 (CH), 56.5 (CH_2_), 54.4 (CH), 51.7 (CH_2_), 51.2 (CH), 41.4 (CH_3_), 20.8 (CH_2_), 8.8 (CH_3_); EIMS *m*/*z* (%) 378 (M+, 5), 220 (20), 219 (87), 218 (100), 204 (26), 176 (11); HR-EI-MS *m*/*z* 378.1692 (M+, calcd for C_21_H_22_N_4_O_3_, 378.1693). The ^1^H-NMR and ^13^C-NMR figures are shown in Appendix A.

To a solution of phenol, **3c** (8.2 mg, 19.8 µmol), in THF (0.5 mL) was added salcomine (8.2 mg, 19.8 µmol, 1.0 eq.) at room temperature, and the reaction mixture was stirred for 1.5 h under an O_2_ atmosphere (Figure 3). The reaction mixture was filtered through a cellulose pad and washed with EtOAc. The filtrate was concentrated in vacuo to give a residue. The residue was purified by SiO_2_ flash column chromatography (CH_2_Cl_2_: MeOH = 99:1) to afford compound 4c (9.8 mg, quant.) as a yellow amorphous substance. [α]^24^_D_ +37.2 (c 0.16, CHCl_3_); IR (KBr) 3017, 2941, 2829, 1652, 1616, 1448, 1307, 1235, 1152, 950, 863, 822, 755, 476, 463, 437, 430, 419, 412 cm^−1^; ^1^H-NMR (400 MHz, CDCl_3_) δ 7.80-7.77 (1H, m, 1′-Np), 7.74-7.71 (2H, m, 1′-Np), 7.64 (1H, s, 1′-Np), 7.50-7.44 (2H, m, 1′-Np), 7.22 (1H, dd, J = 8.4, 1.7 Hz, 1′-Np), 3.95 (3H, s, 9-OCH3), 3.89 (1H, s, 1-H), 3.81 (1H, d, J = 13.3 Hz, 1′-H), 3.71 (1H, d, J = 13.3 Hz, 1′-H), 3.57 (1H, d, J = 2.3 Hz, 4-H), 3.26 (1H, d, J = 7.3 Hz, 5-H), 3.01 (1H, dd, J = 11.7, 3.2 Hz, 2-H), 2.68 (1H, dd, J = 20.6, 7.3 Hz, 6-H), 2.62 (1H, dd, J = 11.7, 0.9 Hz, 2-H), 2.33 (3H, s, 11N-CH_3_), 2.13 (1H, d, J = 20.6 Hz, 6-H), 2.05 (3H, s, 8-CH_3_); ^13^C-NMR (100 MHz, CDCl_3_) δ 186.9 (C), 182.3 (C), 155.4 (C), 141.0 (C), 137.3 (C), 133.7 (C), 133.2 (C), 133.0 (C), 128.6 (CH), 128.6 (C), 127.7 (CH), 127.7 (CH), 127.6 (CH), 126.4 (CH), 126.1 (CH), 126.0 (CH), 115.8 (C), 60.9 (CH_3_), 59.0 (CH_2_), 57.8 (CH), 54.5 (CH), 51.6 (CH_2_), 51.3 (CH), 41.5 (CH_3_), 20.8 (CH_2_), 8.8 (CH_3_); EIMS *m*/*z* (%) 427 (M+, 17), 220 (25), 219 (77), 218 (100), 204 (17), 141 (40); HRMS (EI) *m*/*z* 427.1898 (M+, calcd for C_26_H_25_N_3_O_3_, 427.1896). The ^1^H-NMR and ^13^C-NMR figures are shown in Appendix A.

To a solution of lactam (46.4 mg, 119 µmol) in THF (2.3 mL) at 0 °C was slowly added LiAlH_2_(OEt)_2_ (1.0 mol/L in CH_2_Cl_2_, 1.4 mL, 1.4 mmol, 12 eq.) over 10 min. The reaction mixture was stirred at 0 °C for 3 h. The reaction mixture was quenched with AcOH (145 µL, 2.50 mmol, 21 equiv.), followed by the addition of KCN (55.2 mg, 713 µmol, 6.0 equiv.) in H_2_O (250 µL), and stirring was continued for 15 h at 25 °C (Figure 4). The reaction mixture was neutralized with 5% NaHCO_3_ solution and diluted with saturated Rochell’s salt aq., and the mixture was stirred for 1 h. The reaction mixture was extracted with CHCl_3_ (3 × 50 mL). The combined extracts were washed with brine (50 mL), dried over Na_2_SO_4_, and concentrated in vacuo to give a residue. The residue was purified by SiO_2_ flash column chromatography (CHCl_3_: MeOH = 19:1) to afford compound 2d (34.6 mg, 73%) as a colorless amorphous substance. [α]^24^_D_ −47.0 (c 0.14, CHCl_3_); IR (KBr) 3306, 3012, 2938, 2821, 1483, 1456, 1412, 1322, 1230, 1158, 1061, 1029, 900, 796, 753, 700, 666, 482, 439, 415 cm^−1^; ^1^H-NMR (400 MHz, CDCl_3_) δ 7.39-7.28 (5H, m, 10-OCH_2_Ph-H), 6.65 (1H, s, 7-H), 5.08 (2H, d, J = 2.1 Hz, 10-OCH_2_Ph), 4.10 (1H, d, J = 2.3 Hz, 4-H), 3.92 (1H, s, 1-H), 3.85 (3H, s, 9-OCH_3_), 3.30 (1H, d, J = 8.4 Hz, 5-H), 3.15 (2H, d, J = 2.5 Hz, 1′-H), 3.03 (1H, dd, J = 17.7, 8.4 Hz, 6-H), 2.80 (1H, dd, J = 10.9, 3.2 Hz, 2-H), 2.56 (1H, ddd, J = 10.9, 2.2, 1.1 Hz, 2-H), 2.45 (1H, d, J = 17.7 Hz, 6-H), 2.27 (3H, s, 8-CH_3_), 2.20 (1H, t, J = 2.5 Hz, 3′-H), 2.10 (3H, s, 11N-CH_3_); ^13^C-NMR (100 MHz, CDCl_3_) δ: 148.9 (C), 148.2 (C), 137.4 (C), 130.4 (C), 129.8 (C), 128.6 (CH×2), 128.5 (CH×2), 128.2 (CH), 126.1 (C), 124.6 (CH), 116.0 (C), 78.1 (CH), 74.4 (CH_2_), 73.8 (C), 60.0 (CH_3_), 58.1 (CH), 55.2 (CH), 54.1 (CH_2_), 52.4 (CH), 44.4 (CH_2_), 41.2 (CH_3_), 24.9 (CH_2_), 15.8 (CH_3_); EIMS *m*/*z* (%) 401 (M+, 2), 295 (26), 294 (100), 243 (16), 204 (25), 203 (23); HRMS (EI) *m*/*z* 401.2106 (M+, calcd for C_25_H_27_N_3_O_2_, 401.2103). The ^1^H-NMR and ^13^C-NMR figures are shown in Appendix A. 

To a solution of **2d** (24.0 mg, 59.8 μmol) and pentamethylbenzene (89.0 mg, 598 μmol, 10.0 eq.) in CH_2_Cl_2_ (7.6 mL) was added BCl_3_ (1.0 mol/L in CH_2_Cl_2_, 300 µL, 300 µmol, 5 eq.) over 30 min at −78 °C, and the mixture was stirred for 2 h (Figure 5). The reaction mixture was diluted with CH_2_Cl_2_ (30.0 mL) and quenched with saturated NaHCO_3_ solution at 0 °C. The mixture was extracted with CH_2_Cl_2_ (3 × 40 mL). The combined extracts were washed with brine (30 mL), dried over Na_2_SO_4_, and concentrated in vacuo to give a residue. The residue was purified by SiO_2_ flash column chromatography (n-hexane: AcOEt = 2:1) to afford compound 3d (12.1 mg, 65%) as a colorless amorphous substance. [α]^24^_D_ −78.4 (c 0.09, CHCl_3_); IR (KBr) 3530, 3426, 3307, 3020, 2938, 2825, 1718, 1587, 1458, 1231, 1157, 1059, 756, 668, 503, 458, 434, 428, 422, 404 cm^−1^; ^1^H-NMR (500 MHz, CDCl_3_) δ 6.44 (1H, s, 7-H), 5.63 (1H, brs, 10-OH), 4.15 (1H, d, J = 1.7 Hz, 4-H), 4.08 (1H, s, 1-H), 3.78 (3H, s, 9-OCH_3_), 3.36 (1H, d, J = 7.8 Hz, 5-H), 3.22 (2H, d, J = 2.6 Hz, 1′-H), 3.08 (1H, dd, J = 17.8, 7.8 Hz, 6-H), 2.90 (1H, dd, J = 10.7, 2.6 Hz, 2-H), 2.74 (1H, dt, J = 10.7, 1.1 Hz, 2-H), 2.47 (1H, d, J = 17.8 Hz, 6-H), 2.34 (3H, s, 11N-CH_3_), 2.26 (3H, s, 8-CH_3_), 2.22 (1H, t, J = 2.6 Hz, 3′-H); ^13^C-NMR (125 MHz, CDCl_3_) δ 145.4 (C), 142.8 (C), 130.4 (C), 128.4 (C), 120.7 (CH), 118.9 (C), 116.0 (C), 78.1 (C), 73.8 (CH), 60.7 (CH_3_), 58.1 (CH), 55.3 (CH), 53.4 (CH_2_), 52.0 (CH), 44.5 (CH_2_), 41.5 (CH_3_), 24,9 (CH_2_), 15.8 (CH_3_); EIMS *m*/*z* (%) 311 (M+, 2), 205 (20), 204 (100); HRMS (EI) *m*/*z* 311.1632 (M+, calcd for C_18_H_21_N_3_O_2_, 311.1634). The ^1^H-NMR and ^13^C-NMR figures are shown in Appendix A. 

To a solution of phenol, **3d** (6.0 mg, 19.3 µmol), in THF (0.5 mL) was added salcomine (7.4 mg, 19.3 µmol, 1.0 eq.) at room temperature, and the reaction mixture was stirred for 1.5 h under an O_2_ atmosphere (Figure 6). The reaction mixture was filtered through a cellulose pad and washed with EtOAc. The filtrate was concentrated in vacuo to give a residue. The residue was purified by SiO_2_ flash column chromatography (CH_2_Cl_2_: MeOH = 99:1) to afford compound 4d (5.7 mg, 90%) as a yellow amorphous substance. [α]^24^_D_ −65.2 (c 0.09, CHCl3); IR (KBr) 3307, 3020, 2941, 2853, 1653, 1615, 1458, 1308, 1236, 1216, 1157, 863, 752, 668, 469, 462, 448, 440, 432, 425, 419, 413 cm^−1^; ^1^H-NMR (400 MHz, CDCl_3_) δ 4.05 (1H, d, ¬J = 2.1 Hz, 4-H), 4.02 (3H, s, 9-OCH_3_), 3.85 (1H, d, J = 1.1 Hz, 1-H), 3.37 (1H, dt, J = 7.4, 2.1 Hz, 5-H), 3.25 (2H, d, J = 2.5 Hz, 1′-H), 2.87 (1H, dd, J = 11.4, 3.3 Hz, 2-H), 2.75 (1H, dd, J = 20.8, 7.4 Hz, 6-H), 2.60 (1H, ddd, J = 11.4, 2.1, 1.1 Hz, 2-H), 2.32 (3H, s, 11N-CH_3_), 2.26 (1H, t, J = 2.5 Hz, 3′-H), 2.22 (1H, d, J = 20.8 Hz, 6-H), 1.97 (3H, s, 8-CH_3_); ^13^C-NMR (100 MHz, CDCl_3_) δ 186.9 (C), 182.4 (C), 155.3 (C), 141.0 (C), 136.8 (C), 129.0 (C), 115.4 (C), 77.7 (CH), 74.2 (C), 61.0 (CH_3_), 57.5 (CH), 54.5 (CH), 51.6 (CH_2_), 51.0 (CH), 44.3 (CH_2_), 41.5 (CH_3_), 20.8 (CH_2_), 8.8 (CH_3_); EIMS *m*/*z* (%) 325 (M+, 6), 220 (12), 219 (67), 218 (100), 204 (31); HRMS (EI) *m*/*z* 325.1428 (M+, calcd for C_18_H_19_N_3_O_3_, 325.1426). The ^1^H-NMR and ^13^C-NMR figures are shown in Appendix A.

To a solution of lactam (122 mg, 308 µmol) in THF (8.6 mL) at −78 °C was slowly added DIBAL (1.0 mol/L in toluene, 3.7 mL, 3.7 mmol, 12 eq.) over 10 min. The reaction mixture was stirred at −78 °C for 3 h. The reaction mixture was quenched with MeOH (263 µL, 6.47 mmol, 21 equiv.) and warmed to 0 °C, followed by the addition of AcOH (370 µL, 6.47 mmol, 21 equiv.) and KCN (127 mg, 1.85 mmol, 6.0 equiv.) in H_2_O (650 µL), and stirring was continued for 15 h at 25 °C (Figure 7). The reaction mixture was neutralized with 5% NaHCO_3_ solution and diluted with saturated Rochell’s salt aq., and the mixture was stirred for 1 h. The reaction mixture was extracted with CHCl_3_ (3 × 100 mL). The combined extracts were washed with brine (100 mL), dried over Na_2_SO_4_, and concentrated in vacuo to give a residue. The residue was purified by SiO_2_ flash column chromatography (CHCl_3_: MeOH = 49:1) to afford compound 2e (105 mg, 84%) as a pale orange amorphous substance. [α]^24^_D_ −8.7 (c 1.5, CHCl_3_); IR (KBr) 3485, 2937, 1725, 1580, 1483, 1415, 1321, 1230, 1161, 1067, 902, 754, 700, 667, 473 cm^−1^; ^1^H-NMR (400 MHz, CDCl_3_) δ 7.38-7.30 (5H, m, 10-OCH_2_Ph-H), 6.64 (1H, s, 7-H), 5.09 (2H, s, 10-OCH_2_Ph), 3.91 (1H, s, 1-H), 3.83 (3H, s, 9-OCH_3_), 3.77 (1H, s, 4-H), 3.37 (2H, s, 2′-H), 3.27 (1H, d, J = 7.2 Hz, 5-H), 3.07 (1H, dd, J = 17.7, 7.3 Hz, 6-H), 2.87 (1H, dd, J = 11.0, 3.0 Hz, 2-H), 2.63-2.49 (3H, m, 2-H, 1′-H, overlapped), 2.40 (1H, d, J = 17.7 Hz, 6-H), 2.25 (3H, s, 8-CH_3_), 2.14 (3H, s, 11N-CH_3_); ^13^C-NMR (100 MHz, CDCl_3_) δ: 149.2 (C), 148.1 (C), 137.4 (C), 130.7 (C), 129.5 (C), 128.6 (CH×2), 128.5 (CH×2), 128.2 (CH), 126.1 (C), 124.4 (d, C-7), 116.5 (C), 74.6 (CH_2_), 60.1 (CH_3_), 59.8 (CH), 57.4 (CH_2_), 55.9 (CH_2_), 55.4 (CH), 52.8 (CH), 52.6 (CH_2_), 41.3 (CH_3_), 25.0 (CH_2_), 15.8 (CH_3_); EIMS *m*/*z* (%) 407 (M+, 0.2), 295 (23), 294 (100), 204 (18), 203 (35); HRMS *m*/*z*: 407.2213 (M+, calcd for C_24_H_29_N_3_O_3_, 407.2209). The ^1^H-NMR and ^13^C-NMR figures are shown in Appendix A.

To a solution of **2e** (58.9 mg, 145 μmol) and pentamethylbenzene (216 mg, 1.45 mmol, 10.0 eq.) in CH_2_Cl_2_ (18 mL) was added BCl_3_ (1.0 mol/L in CH_2_Cl_2_, 723 µL, 723 µmol, 5 eq.) over 30 min at −78 °C and the mixture was stirred for 2 h (Figure 8). The reaction mixture was diluted with CH_2_Cl_2_ (30.0 mL) and quenched with saturated NaHCO_3_ solution at 0 °C. The mixture was extracted with CH_2_Cl_2_ (3 × 100 mL). The combined extracts were washed with brine (100 mL), dried over Na_2_SO_4_, and concentrated in vacuo to give a residue. The residue was purified by SiO_2_ flash column chromatography (n-hexane: AcOEt = 1:2) to afford compound **3e** (31.2 mg, 68%) as a pale orange amorphous substance. [α]^24^_D_ −59.6 (c 0.23, CHCl_3_); IR (KBr) 3528, 3020, 2942, 2399, 1502, 1457, 1417, 1367, 1329, 1215, 1161, 1059, 1024, 748, 669, 490, 451, 418, 411, 404 cm^−1^; ^1^H-NMR (500 MHz, CDCl_3_) δ 6.43 (1H, s, 7-H), 5.65 (1H, brs, 10-OH), 4.10 (1H, s, 1-H), 3.81 (1H, s, 4-H), 3.77 (3H, s, 9-OCH_3_), 3.40 (2H, dd, J = 10.6, 5.2 Hz, 2′-H), 3.33 (1H, d, J = 7.4 Hz, 5-H), 3.12 (1H, dd, J = 18.0, 7.4 Hz, 6-H), 2.98 (1H, dd, J = 11.1, 2.7 Hz, 2-H), 2.70 (1H, dt, J = 11.1, 1.1 Hz, 2-H), 2.68-2.56 (2H, m, 1′-H), 2.42 (1H, d, J = 18.0 Hz, 6-H), 2.39 (3H, s, 11N-CH_3_), 2.24 (3H, s, 8-CH_3_); ^13^C-NMR (125 MHz, CDCl_3_) δ 145.3 (C), 143.1 (C), 130.1 (C), 128.6 (C), 120.5 (CH), 118.9 (C), 116.7 (C), 60.8 (CH_3_), 59.7 (d, C-4), 57.4 (CH_2_), 56.0 (CH_2_), 55.5 (CH), 52.4 (CH), 52.0 (CH_2_), 41.6 (CH_3_), 25.1 (CH_2_), 15.8 (CH_3_); EIMS *m*/*z* (%) 317 (M+, 0.3), 234 (10), 205 (18), 204 (100), 189 (12); HRMS (EI) *m*/*z* 317.1735 (M+, calcd for C_17_H_23_N_3_O_3_, 317.1739). The ^1^H-NMR and ^13^C-NMR figures are shown in Appendix A. 

To a solution of phenol, **3e** (21.5 mg, 67.7 µmol), in THF (1.8 mL) was added salcomine (25.2 mg, 67.7 µmol, 1.0 eq.) at room temperature, and the reaction mixture was stirred for 1 h under an O_2_ atmosphere (Figure 9). The reaction mixture was filtered through a cellulose pad and washed with EtOAc. The filtrate was concentrated in vacuo to give a residue. The residue was purified by SiO2 flash column chromatography (CH_2_Cl_2_: MeOH = 99:1) to afford compound **4e** (11.2 mg, 50%) as a yellow amorphous substance. [α]^24^_D_ −28.7 (c 0.12, CHCl_3_); IR (KBr) 3506, 2944, 2849, 1653, 1308, 1236, 1160, 955, 866, 755, 603, 459, 436, 422, 415, 406 cm^−1^; ^1^H-NMR (500 MHz, CDCl_3_) δ 4.03 (3H, s, 9-OCH_3_), 3.87 (1H, d, J = 1.1 Hz, 1-H), 3.83 (1H, d, J = 2.1 Hz, 4-H), 3.61–3.56 (2H, m, 2′-H), 3.36 (1H, dt, J = 7.4, 2.1 Hz, 5-H), 2.94 (1H, dd, J = 11.6, 3.3 Hz, 2-H), 2.77 (1H, dd, J = 20.9, 7.4 Hz, 6-H), 2.73–2.61 (2H, m, 1′-H), 2.59 (1H, ddd, J = 11.6, 2.3, 1.1 Hz, 2-H), 2.33 (3H, s, 11N-CH_3_), 2.23 (1H, d, J = 20.9 Hz, 6-H), 1.96 (3H, s, 8-CH_3_); ^13^C-NMR (125 MHz, CDCl_3_) δ 186.6 (C), 182.3 (C), 155.3 (C), 140.9 (C), 137.2 (C), 128.8 (C), 115.9 (C), 61.0 (CH_3_), 59.1 (CH), 58.5 (CH_2_), 56.4 (CH_2_), 54.5 (CH), 51.3 (CH_2_), 51.2 (CH), 41.5 (CH_3_), 20.9 (CH_2_), 8.8 (CH_3_); EIMS *m*/*z* (%) 331 (M+, 11), 305 (11), 304 (57), 232 (11), 221 (13), 220 (94), 219 (93), 218 (100), 205 (17), 204 (61), 202 (27), 201 (15), 190 (26), 176 (20), 86 (17), 85 (15), 42 (12); HRMS (EI) *m*/*z*: 331.1529 (M+, calcd for C_17_H_21_N_3_O_4_, 331.1532). The ^1^H-NMR and ^13^C-NMR figures are shown in Appendix A. 

To a solution of p-quinone (3.7 mg, 11.2 µmol) in THF (0.5 mL) was added phthalimide (8.5 mg, 55.8 µmol, 5.0 equiv.), triphenylphosphine (15.9 mg, 55.8 µmol, 5.0 equiv.), and diethyl azodicarboxylate (2.2 mol/L in toluene, 25 µL, 55.8 µmol, 5.0 equiv.) at room temperature, and the reaction mixture was stirred for 3 h under an Ar atmosphere (Figure 10). The solvent was removed in vacuo, and the residue was purified by SiO_2_ flash column chromatography (CH_2_Cl_2_: MeOH = 99:1) to afford compound 4f (2.8 mg, 55%) as a yellow oil. [α]^24^_D_ +97.1 (c 0.04, CHCl_3_); IR (KBr) 3025, 2947, 2853, 1771, 1450, 1238, 1109, 953, 922, 866, 807, 719, 666, 529, 469, 462, 453, 445, 426, 414 cm^−1^; ^1^H-NMR (500 MHz, CDCl_3_) δ 7.73–7.64 (4H, m, 5′-H, 6′-H, 7′-H, 8′-H, overlapped), 4.08 (3H, s, 9-OCH_3_), 3.83 (1H, s, 1-H), 3.81–3.75 (1H, m, 2′-H), 3.79 (1H, d, J = 2.3 Hz, 4-H), 3.69-3.64 (1H, m, 2′-H), 3.26 (1H, d, J = 7.2 Hz, 5-H), 2.91–2.85 (1H, m, 1′-H), 2.80 (2H, d, J = 2.9 Hz, 2-H), 2.78–2.74 (1H, m, 1′-H), 2.54 (1H, dd, J = 20.7, 7.2 Hz, 6-H), 2.24 (3H, s, 11N-CH_3_), 1.99 (1H, d, J = 20.7 Hz, 6-H), 1.77 (3H, s, 8-CH_3_); ^13^C-NMR (125 MHz, CDCl_3_) δ 186.2 (C), 182.0 (C), 168.1 (C×2), 155.5 (C), 140.0 (C), 136.9 (C), 134.0 (CH×2), 131.4 (C×2), 127.2 (C), 123.3 (CH×2), 115.9 (C), 60.9 (CH_3_), 59.3 (CH), 5.45 (CH), 52.9 (CH_2_), 50.9 (CH), 50.6 (CH_2_), 41.3 (CH_3_), 34.3 (CH_2_), 20.3 (CH_2_), 8.7 (CH_3_); EIMS *m*/*z* (%) 460 (M+, 15), 117 (18), 220 (27), 219 (100), 218 (74), 204 (22), 176 (12), 174 (17); HRMS *m*/*z* 460.1744 (M+, calcd for C_25_H_24_N_4_O_5_, 460.1747). The ^1^H-NMR and ^13^C-NMR figures are shown in Appendix A.

### 4.4. Preparation of Stock Solution for RT Derivatives

The RT analogs for the right half (DH_18, DH_21, DH_32, DH_35, DH_38, and DH_39) were prepared as 50 mM stock solutions by dissolving them in dimethyl sulfoxide (DMSO) and subsequently stored at −20 °C. In the high treatment conditions, the DMSO was present at a final concentration of 0.2% v/v, and no cytotoxic effects were observed.

### 4.5. Cell Viability Assay

The cell viability was assessed using an MTT assay, determining the half-maximal inhibitory concentration (IC_50_) to screen right-half RT analogs (DH_18, DH_21, DH_32, DH_35, DH_38, and DH_39). Human NSCLC cells (A549, H23, and H292) were seeded in a 96-well cell culture plate at a density of 1 × 10^4^ cells per well and incubated for 24 h at 37 °C. Subsequently, the culture medium was aspirated, and a complete medium with varying concentrations (0–100 μM) of right-half RT analogs was added for 24 h. Following the removal of the compound-containing medium, an MTT reagent at a concentration of 4 mg/mL was applied and incubated for 3 h at 37 °C. The MTT reagent was replaced with 100 μL of DMSO to dissolve the formazan crystals. The measurement of the formazan product was conducted at 570 nm utilizing a microplate reader. RT served as the positive control in this experiment.

### 4.6. Nuclear Staining Assay

The assessment of apoptosis and dead cell counts was performed through costaining with Hoechst 33342 and propidium iodide (PI). Human NSCLCs (1 × 10^4^ cells per well) were seeded in a 96-well plate, incubated at 37 °C, and exposed to all right-half RT analogs at a concentration of 0.5 μM. After treatment, the cells were rinsed with PBS and subjected to co-staining with Hoechst 33342 (10 μg/mL) and propidium iodide (PI) (0.02 μg/mL) for 30 min at 37 °C. Fluorescence microscopy (Olympus 1X51 with a DP70 digital camera, Tokyo, Japan) was utilized to observe the fragmented nuclei indicative of apoptotic cells stained with Hoechst 33342 and the positive PI staining for dead cells. Subsequently, the percentages of apoptosis and dead cells were calculated. In this experiment, RT at a concentration of 0.5 μM served as the positive control.

### 4.7. Proliferation Assay

An MTT assay was employed to evaluate the antiproliferative impact of the right-half RT analog, DH_32, over a period of three consecutive days. The human NSCLC cells lines A549, H23, and H292 were seeded at a cell density of 2 × 10^3^ cells per well in a 96-well plate. The cells were exposed to low concentrations (0, 25, 50, and 100 nM) of DH_32 at various time points (0 h, 24 h, 48 h, and 72 h) and then incubated under 5% CO_2_ at 37 °C. Subsequently, the cell viability was assessed using an MTT assay at 0 h, 24 h, 48 h, and 72 h. The percentage of cell proliferation was determined by dividing the optical density (OD) of the cells at each time point by the OD of the non-treated control cells. In this experiment, RT served as the positive control.

### 4.8. Colony Formation Assay

A clonogenic or colony formation assay was utilized to explore the capability of cancer cells to proliferate from single cells and form tumor colonies in response to DH_32 treatment. A single-cell suspension of NSCLC cells (A549, H23, and H292) was seeded at a density of 250 cells per well in a 6-well plate and permitted to adhere for 12 h. The cells were treated to different concentrations of DH_32 (0, 25, 50, and 100 nM) for a duration of 24 h. Subsequently, the medium containing the drug was aspirated, and fresh complete medium was introduced. The colonies were cultured for a period of 7 days, with the medium being renewed every 2 days. The colonies were washed with PBS, followed by fixation using a solution of methanol and acetic acid (3:1 *v*/*v*) for 5 min. Subsequently, they were stained with crystal violet (0.05% *w*/*v* in 4% paraformaldehyde) for 30 min. The excess crystal violet stain was washed away with distilled water until a clear background was obtained, and the samples were left to air-dry at room temperature. The colonies were photographed, and the captured images were analyzed using ImageJ software to determine the count of stained colonies. In this experiment, RT served as the positive control.

### 4.9. Three-Dimensional (3D) CSC Spheroid Formation

A three-dimensional (3D) spheroid was employed to generate the spheroids containing CSC subpopulations from lung cancer cells. For CSC-rich population establishment, the spheroid culture assay used in this study was slightly modified from a previously described method [45,60]. Lung cancer cells, specifically NSCLC cells (A549, H23, and H292), were seeded onto a 24-well ultra-low attachment plate, with approximately density of 2.5 × 10^3^ cells per well in a medium with a 1% FBS and Nutristem mix. The primary spheroids were allowed to form for 7 days. Every 2 days, fresh medium with 1% FBS and Nutristem mix was replenished. On day 7 of primary spheroid culture, primary spheroids were resuspended into single cells using trypsin, and again 2.5 × 10^3^ cells/well were seeded into a 24-well ultra-low attachment plate. Secondary spheroids were allowed to form for 10 days. After 10 days, the secondary spheroids were treated with 50 nM concentration DH_32 for 24 h treatment. The spheroids that had undergone treatment were retrieved through centrifugation at 1000× *g* for 30 s and then rinsed with PBS. The spheroids were fixed with 4% paraformaldehyde in a 50 μL PBS solution for 20 min at room temperature, and then they were made permeable by treating them with 0.1% Triton X in PBS for 5 min at room temperature. The spheroids were exposed to a mixture of 5% FBS and 2% BSA in PBS for 1 h at room temperature to prevent non-specific protein interactions. Subsequently, the spheroids were incubated with primary antibodies (CD133, CD44, and OCT4) at a 1:250 dilution in a 30 μL PBS solution containing 2.5% FBS and 1% BSA, and this incubation took place overnight at 4 °C. On the subsequent day, the spheroids were rinsed with PBS and then exposed to secondary antibodies (Alexa Fluor 594-conjugated goat anti-rabbit IgG (H+L) and Alexa Fluor 488-conjugated goat anti-mouse IgG (H+L)) at a 1:500 dilution in a 30 μL PBS solution containing 2.5% FBS and 1% BSA at room temperature for 1 h 30 min. These spheroids were subjected to staining with Hoechst 33342 (Sigma, St. Louis, MO, USA) at a concentration of 10 μg/mL for a duration of 15 min at room temperature. Subsequently, the fluorescence signals were observed using a fluorescence microscope (Olympus IX 51 with DP70; Olympus America Inc., Center Valley, PA, USA). The intensity of fluorescence was quantified using Image J software. NSCLC spheroids treated with RT at 50 nM and cisplatin at 50 nM served as the positive control.

### 4.10. Reverse Transcription Quantitative Polymerase Chain Reaction (RT-qPCR)

RNA extraction was performed on DH_32-treated cells (3 × 10^5^ cells per well in 6-well plates) using GENEzol as the reagent. Total RNA was used to synthesize cDNA through the application of SuperScript III reverse transcriptase from Invitrogen. Following cDNA synthesis, 100 ng of the cDNA was utilized for RT-qPCR, employing Luna Universal qPCR Master Mix from NEB, UK, in a final volume of 20 µL. The reaction mixture was conducted using the CFX 96 real-time PCR system from Bio-Rad, located in Hercules, CA. For the RT-qPCR, the protocol included an initial denaturation step at 95 °C for 1 min, followed by 45 cycles involving denaturation at 95 °C for 10 s, annealing and extension at 60 °C for 30 s. Melting curve analysis was employed to assess the specificity of the primers. The targeted primer genes were:

OCT4 (Fwd) TCGAGAACCGAGTGAGAGG, Tm = 58.8 °C 

OCT4 (Rev) GAACCACACTCGGACCACA, Tm = 58.8 °C

NANOG (Fwd) ATGCCTCACACGGAGACTGT, Tm = 59.4 °C 

NANOG (Rev) AAGTGGGTTGTTTGCCTTTG, Tm = 58 °C 

SOX2 (Fwd) TGATGGAGACGGAGCTGAA, Tm = 56.7 °C 

SOX2 (Rev) GGGCTGTTTTTCTGGTTGC Tm = 56.7 °C

GAPDH (Fwd) CCACCCATGGCAAATTCCATGGCA Tm = 67 °C 

GAPDH (Rev) TCTAGACGGCAGGTCAGGTCCACC Tm = 70.4 °C

The gene expression levels were normalized using the GAPDH gene as an internal control. The relative mRNA gene expression level for each gene was determined based on the comparative Cq values. RT acted as the positive control for this experiment.

### 4.11. Immunofluorescence

A549, H23, and H292 NSCLC cells were seeded at a concentration of 1 × 10^4^ cells per well in 96-well plates. The cells were exposed to different concentrations of DH_32 (0, 25, 50, and 100 nM) for 24 h, after which the medium was removed. Following that, the cells were subjected to fixation with 4% paraformaldehyde for 15 min. The cells were permeabilization with 0.5% Triton-X for 5 min, followed by the blocking of non-specific proteins with 10% FBS in 0.1% Triton-X PBS for 1 h at room temperature. The cells were exposed to a 1:400 dilution of primary antibodies (CD133, CD44, ALDH1A1, OCT4, NANOG, SOX2, and β-catenin) and incubated at 4 °C overnight. On the subsequent day, the cells were treated with a 1:500 dilution of the following secondary antibodies: Alexa Fluor 594-conjugated goat anti-rabbit IgG (H+L) (Invitrogen) and Alexa Fluor 488-conjugated goat anti-mouse IgG (H+L) (Invitrogen). Simultaneously, the localization of nuclei was stained with Hoechst 33342 for 1 h at room temperature. The cells were rinsed with PBS and then covered with 50% glycerol. The images were captured by a fluorescence microscope (Olympus IX 51 with DP70; Olympus America Inc., Center valley, PA, USA). The fluorescence intensity was measured by Image J software. RT acted as the positive control for this experiment.

### 4.12. Western Blot Analysis

The protein expression levels in NSCLC cells were assessed through Western blot analysis. A549, H23, and H292 cells were seeded at a density of 4 × 10^5^ cells per well in 6-well cell culture plates. These cells were subjected to treatments with DH_32 at concentrations of 0, 25, 50, and 100 nM for 24 h at 37 °C.

Following treatment, the cells were collected using cold PBS and incubated on ice with a lysis buffer. The lysis buffer contained 50 mM 4-(2-hydroxyethyl)-1-piperazineethanesulfonic acid (pH 7.5), 150 mM NaCl, 5 mM EDTA, 1% Triton X-100, 1 mM phenylmethylsufonyl fluoride, and 2 μg/mL pepstatin A (cat no: 9803, Cell Signaling), along with complete protease inhibitor cocktail tablets provided in EASYpack (Roche, cat no: 04693116001). The incubation occurred for 40 min on ice, followed by centrifugation at 12,000× *g* at 4 °C for 15 min. Cell lysates were collected, and the protein content was quantified using the bicinchoninic acid (BCA) protein kit from Thermo-Fisher Scientific, Rockford, IL, USA. The sample was denatured by heating at 95 °C for 5 min with a sample buffer. The protein was loaded onto a 10% sodium dodecyl sulfate polyacrylamide gel electrophoresis (SDS-PAGE). Following gel separation, the proteins were transferred onto a nitrocellulose membrane with a pore size of 0.45 μm. The membranes were incubated with 5% nonfat dry milk in TBST (25 mm Tris-HCl, pH 7.4, 125 mm NaCl, and 0.05% Tween 20) for 1 h 30 min. Following this, the membrane was exposed to primary antibodies (Cell Signaling) including rabbit PARP, BcL2, Bax, CD133, CD44, OCT4, ALDH1A1, and β-catenin, and incubated at 4 °C overnight. Rabbit β-actin served as the control for loading normalization. The primary antibodies were diluted to 1:1000 in 5% *w*/*v* BSA in TBST. The membranes were washed with TBST and were then exposed to horseradish-peroxidase (HRP)-conjugated isotype-specific secondary antibodies, as well as anti-rabbit and anti-mouse IgG (1:2000 in 5% *w*/*v* skim milk in TBST), for 1 h at RT. Membranes were washed again with TBST. A chemiluminescent substrate (Supersignal West Pico; Pierce, Rockford, IL, USA) was applied to detect complex reactivity, and the signal intensity was quantified through densitometry. Protein intensity was analyzed by using ImageJ software (Image J 1.52a, Rasband, W., National Institutes of Health, Bethesda, MD, USA). RT acted as the positive control for this experiment.

### 4.13. Immunoprecipitation

The DH_32-treated A549, H23, and H292 cells were collected using cold PBS and incubated on ice in a lysis buffer. This buffer consisted of 50 mM 4-(2-hydroxyethyl)-1-piperazineethanesulfonic acid at pH 7.5, 150 mM NaCl, 5 mM EDTA, 1% Triton X-100, 1 mM phenylmethylsulfonyl fluoride, and 2 μg/mL pepstatin A (catalog no: 9803, Cell Signaling), along with complete protease inhibitor cocktail tablets from the EASYpack (Roche, catalog no: 04693116001). The incubation took place for 40 min on ice, followed by centrifugation at 12,000× *g* at 4 °C for 15 min. 

Subsequently, immunoprecipitation was carried out using the Dynabeads™ Protein G Immunoprecipitation Kit from Thermo Fisher Scientific Inc. (Waltham, MA, USA). The magnetic beads were prepared and added to 100 μL of Ab binding and washing buffer that contained β-catenin (2 μL) for 15 min on rotation at room temperature. The complex of magnetic beads and antibodies was re-suspended in a protein lysis buffer and left to incubate at 4 °C overnight, facilitating the binding of the β-catenin antigen with the magnetic bead–antibody complex. The magnetic bead–Ab–Ag complex was washed three times using 200 μL of washing buffer. The magnetic bead–antibody–antigen complex was resuspended in the lysis buffer (30 μL), and the 6x sample buffer (5 μL) was denatured by heating at 95 °C for 5 min. The antigen–antibody complex was loaded onto a 12% sodium dodecyl sulfate polyacrylamide gel electrophoresis (SDS-PAGE). Following gel separation, the proteins were transferred onto a nitrocellulose membrane with a pore size of 0.45 μm. After transfer, the membrane was placed in a mixture containing 5% nonfat dry milk in TBST (composed of 25 mm Tris-HCl, pH 7.4, 125 mm NaCl, and 0.05% Tween 20) for a duration of 1 h 30 min. Following this, the proteins were exposed to primary antibodies (Cell Signaling), specifically ubiquitin mouse mAb (cat no: 14049), and left to incubate overnight at 4 °C. After that, the membrane was exposed to isotype-specific secondary antibodies linked to horseradish peroxidase (HRP), specifically anti-mouse IgG. These secondary antibodies were used at a 1:2000 dilution in a solution containing 5% *w*/*v* skim milk in TBST, and the incubation took place at room temperature for 1 h. The protein bands were identified using an enhanced chemiluminescent detection method, and the bands were then exposed to X-ray film. Protein intensity was analyzed by using ImageJ software (Image J 1.52a, Rasband, W., National Institutes of Health, USA). RT-treated (50 nM) NSCLCs were used as the positive control. 

### 4.14. Molecular Docking

The structures of GSK-3β (PDB: 5K5N) [61] and β-catenin (PDB: 7AFW) [62] were downloaded from the Research Collaboratory for Structural Bioinformatics Protein Data Bank [63]. Missing residues in the protein structures were filled using the loops/refinement model in UCSF Chimera 1.15 [64] with the standard protocol. To prepare DH_32, its 3D conformer was constructed using Marvinsketch and then optimized with the GFN2-xTB method [65] implemented in the XTB 6.5.0 package [66]. 

The Dock 6.9 package [67] was used for all stages of molecular docking. The binding site of GSK-3β has a native ligand known as PF-04802367, which served as a reference to determine the GSK-3β binding site. Similarly, the binding site of β-catenin has a native ligand called R9Q, which was used as a reference to determine the β-catenin binding site. The grid score (kcal/mol) was determined using the standard flexible docking protocol and was calculated as the sum of the van der Waals (VDW) and electrostatic (ES) energies. A grid scores lower than or equal to the reference compound indicates potential interaction with the receptor. Furthermore, 3D visualization and analysis were carried out using UCSF ChimeraX [68].

### 4.15. Statistical Analysis 

The results were reported as the mean ± standard deviation (SD) based on a minimum of three independent biological experiments. Multiple comparisons were conducted using one-way ANOVA analysis with a post hoc test in GraphPad Prism software version 9.0 (GraphPad Software, La Jolla, CA, USA). Statistically significant differences between groups were considered significant at a *p*-value < 0.05.

## 5. Conclusions

This study unveiled that the right-half RT analog, DH_32, inhibits both stem cell markers and transcription factors in cancer-stem-cell-enriched populations in NSCLCs by promoting β-catenin-induced proteasomal degradation. Furthermore, the synthesis of novel derivatives of the right-half of RT, such as DH_32, is simpler and involves fewer steps compared to the synthesis of the RT compound. Moreover, the novel derivative DH_32 exhibited greater potency than the parent RT compound, particularly at a low dose of 25 nM. These results emphasize the potential of DH_32 as a promising compound for targeted anti-cancer strategies in clinical experiments.

## Data Availability

The datasets used and/or analyzed during the current study are available from the corresponding author on reasonable request.

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
