# Peer review of "Simplified Synthesis of Renieramycin T Derivatives to Target Cancer Stem Cells via β-Catenin Proteasomal Degradation in Human Lung Cancer"

_marinedrugs, 2023, doi:10.3390/md21120627_

Round 1

Reviewer 1 Report

Comments and Suggestions for Authors

The article provides a detailed introduction to the superiority of Renieramycin T(RT) Derivatives-DH32 in inhibiting the growth of A549/H23/H239 cells compared to RT, especially emphasizing its ability to target CSCs. It explores how the compound mainly acts on β-catenin, leading to proteasomal degradation and reducing the expression of stem cell markers CD133, CD44, ALDH1A1, and stem cell transcription factors OCT4, NANOG, and SOX2.

Suggestions:

1In the “Introduction” section or the “Results” section, it would be beneficial to provide some explanations on why the A549/H23/H239 cell lines were selected for this study.

2The repeated use of “non-toxic concentrations” in the text may not be precise in terms of determining toxicity. It could be considered to use “low concentrations” instead. Particularly in the Colony Formation Assay, if the compound is at a non-toxic concentration, crystal violet can be used to stain the colony-forming cells (250 cells). Because non-toxic concentrations imply that the compound does not significantly affect the viability and proliferation of the cells.

3The abbreviation “RT” is introduced for “Renieramycins T” in the first occurrence in the main text, but in the subsequent “Discussion” and “Materials and Methods” sections, “Renieramycins T (RT)” is used.

4In the “Materials and Methods” section, it is suggested to use the commonly accepted expressions for RT-PCR, qPCR, and RT-qPCR.

Comments on the Quality of English Language

The abbreviation “RT” is introduced for “Renieramycins T” in the first occurrence in the main text, but in the subsequent “Discussion” and “Materials and Methods” sections, “Renieramycins T (RT)” is used.

Please check the English carefully. There are some grammatical errors.

Author Response

Reviewer 1

Comments and Suggestions for Authors

The article provides a detailed introduction to the superiority of Renieramycin T(RT) Derivatives-DH32 in inhibiting the growth of A549/H23/H239 cells compared to RT, especially emphasizing its ability to target CSCs. It explores how the compound mainly acts on β-catenin, leading to proteasomal degradation and reducing the expression of stem cell markers CD133, CD44, ALDH1A1, and stem cell transcription factors OCT4, NANOG, and SOX2.

Suggestions:

1、In the “Introduction” section or the “Results” section, it would be beneficial to provide some explanations on why the A549/H23/H239 cell lines were selected for this study.

Response: Thank you for your suggestion. In the results section, we explained the rationale for selecting these cell lines.

2、The repeated use of “non-toxic concentrations” in the text may not be precise in terms of determining toxicity. It could be considered to use “low concentrations” instead. Particularly in the Colony Formation Assay, if the compound is at a non-toxic concentration, crystal violet can be used to stain the colony-forming cells (250 cells). Because non-toxic concentrations imply that the compound does not significantly affect the viability and proliferation of the cells.

Response: Thank you for your suggestion. Yes, we agreed with this term to use 'Low concentrations’.

3、The abbreviation “RT” is introduced for “Renieramycins T” in the first occurrence in the main text, but in the subsequent “Discussion” and “Materials and Methods” sections, “Renieramycins T (RT)” is used.

Response: Thank you for your suggestion. We have already revised it.

4、In the “Materials and Methods” section, it is suggested to use the commonly accepted expressions for RT-PCR, qPCR, and RT-qPCR.

Response: Thank you for your suggestion. We used the term reverse transcription quantitative polymerase chain reaction (RT-qPCR) techniques.

Comments on the Quality of English Language

The abbreviation “RT” is introduced for “Renieramycins T” in the first occurrence in the main text, but in the subsequent “Discussion” and “Materials and Methods” sections, “Renieramycins T (RT)” is used.

Please check the English carefully. There are some grammatical errors.

Response: Thank you for your suggestion. We have already revised it.

Reviewer 2 Report

Comments and Suggestions for Authors

The authors developed a simpler chemical synthesis of a variety of derivatives of Renieramycin T (RT), an alkaloid compound isolated from the blue sponge. Cell proliferation and death assays identified one derivative, DH32 as being as potent as parental RT using 3 different lung cancer cell lines. Lower concentrations of DH32 and RT (25, 50, 100 nM) were used to demonstrate inhibition of proliferation using colony assays and by western blot of apoptosis proteins.  Spheroid assays were used to show DH32 and RT lower stem cell markers and beta-catenin. The authors suggest a greater ubiquitination of beta-catenin in DH32 treated cells to promote proteasomal degradation and block stem cell transcription factors.  Molecular docking results suggest DH32 binds to beta-catenin.

Overall, this manuscript appears to have some value, especially in development of the synthesis of RT derivatives. There are some comments and suggestions intended to improve the quality of the manuscript.

1.    Whether DH32 is more potent than parental RT compound is questionable. Fig. 2A IC50 have small differences.  Fig. 3 shows some differences but there are variations between cell lines, so overall I would suggest no major difference. Believe the result is still good but to say DH32 is more potent should be revised throughout the text.

2.    Fig. 2B, for Hoechst stain, add small white arrows pointing to cells with condensed/fragmented nuclei. PI red cells are not necessarily necrotic; more accurate to say “dead” cells.

3.    Evidence to suggest that DH32 or RT increases proteasomal degradation of beta-catenin resulting in decrease of stem cell transcription factors is weak. The result in Fig. 6c is not convincing of a big difference. Perhaps suggest as potential mechanism that needs further investigation.

4.    Oct4 and Nanog forward primers used for qRT-PCR do not match in a Blast search. All other primer sequences are correct. Please clarify.

5.    Most important, there is no mention of potential use of RT or DH32 in animal models of lung cancer. It is critical to determine if RT or DH32 can be used in vivo and whether pharmacokinetics and bioavailability are favorable for use. Otherwise, the value of the work is diminished.

Comments on the Quality of English Language

Some English editing is required.

Author Response

Reviewer 2

Comments and Suggestions for Authors

The authors developed a simpler chemical synthesis of a variety of derivatives of Renieramycin T (RT), an alkaloid compound isolated from the blue sponge. Cell proliferation and death assays identified one derivative, DH32 as being as potent as parental RT using 3 different lung cancer cell lines. Lower concentrations of DH32 and RT (25, 50, 100 nM) were used to demonstrate inhibition of proliferation using colony assays and by western blot of apoptosis proteins.  Spheroid assays were used to show DH32 and RT lower stem cell markers and beta-catenin. The authors suggest a greater ubiquitination of beta-catenin in DH32 treated cells to promote proteasomal degradation and block stem cell transcription factors.  Molecular docking results suggest DH32 binds to beta-catenin.

Overall, this manuscript appears to have some value, especially in development of the synthesis of RT derivatives. There are some comments and suggestions intended to improve the quality of the manuscript.

  1. Whether DH32 is more potent than parental RT compound is questionable. Fig. 2A IC50 have small differences. Fig. 3 shows some differences but there are variations between cell lines, so overall I would suggest no major difference. Believe the result is still good but to say DH32 is more potent should be revised throughout the text.

Response: Thank you for your suggestion. In the contemporary landscape of cancer treatment, the development of various types of anti-cancer drugs has been notable. Two primary categories of anti-cancer therapies are chemotherapeutic drugs (conventional chemotherapy) and targeted therapy (precise cancer treatment that selectively disrupts specific molecules or pathways in cancer cells). Recent research has suggested that targeted therapy offers superior benefits and efficacy compared to traditional chemotherapy (1, 2).

Chemotherapeutic drugs typically undergo screening tests for cytotoxicity, often measured by the IC50 value (3). In contrast, targeted therapy represents an innovative approach to cancer treatment, focusing on the specific modifications in cancer cells responsible for their growth, division, and spread (4).

In part of targeted therapy, cancer stem cell-targeted therapy is an innovative approach focused on eliminating cancer stem cells (CSCs), a subset of cells within tumors that possess self-renewal capabilities and contribute to tumor growth, recurrence, and resistance to traditional treatments such as chemotherapy; this specialized therapy aims to inhibit or disrupt CSC-related pathways or proteins, preventing tumor progression and enhancing overall treatment efficacy (5).

Literature supports the notion that the targeted delivery of therapeutic agents to CSCs holds promise in enhancing the efficacy of cancer treatment and preventing relapse (6). Thus, drugs designed to target CSCs, with a focus on inhibiting or decreasing CSC-related pathways or proteins, are increasingly considered more intriguing than those emphasizing cytotoxic potency (7, 8).

Our findings, as illustrated in Figures 4 and 5, demonstrate that treatment with DH_32 leads to a significant reduction in the levels of CSCs markers, including CD133, CD44, and ALDH1A1, as well as stem cell transcription factors such as OCT4, NANOG, and SOX2 in non-small cell lung cancers (NSCLCs). Therefore, as we aim to investigate the targeted therapy (CSC targeting), the superiority of the DH_32 over RT should be investigated via the inhibition of CSCs-related pathways instead of direct cytotoxicity.

References

  1. Zhong, L.; Li, Y.; Xiong, L.; Wang, W.; Wu, M.; Yuan, T.; et al. Small molecules in targeted cancer therapy: Advances, challenges, and future perspectives. Signal Transduct Target Ther 20216(1), 201.
  2. Viktorsson, K.; Rieckmann, T.; Fleischmann, M.; Diefenhardt, M.; Hehlgans, S.; Rödel, F. Advances in molecular targeted therapies to increase efficacy of (chemo) radiation therapy. Strahlentherapie und Onkologie 2023, 1-19.
  3. Arbabi Moghadam, S.; Rezania, V.;Tuszynski, J.A. Cell death and survival due to cytotoxic exposure modelled as a two-state Ising system. R Soc Open Sci 20207(2), 191578.
  4. Pucci, C.; Martinelli, C.; Ciofani, G. Innovative approaches for cancer treatment: Current perspectives and new challenges. Ecancermedicalscience 2019, 13.
  5. Miyoshi, N.; Haraguchi, N.; Mizushima, T.; Ishii, H.; Yamamoto, H.; Mori, M. Targeting cancer stem cells in refractory cancer. Regen Ther 2021, 17, 13-19.
  6. Shibata, M.; Hoque, M.O. Targeting cancer stem cells: a strategy for effective eradication of cancer. Cancers 2019, 11(5), 732.
  7. Mai, Y.; Su, J.; Yang, C.; Xia, C.; Fu, L. The strategies to cure cancer patients by eradicating cancer stem-like cells. Mol Cancer 2023, 22(1), 171.
  8. Yang, L.; Shi, P.; Zhao, G.; Xu, J.; Peng, W.; Zhang, J.; et al. Targeting cancer stem cell pathways for cancer therapy. Signal Transduct Target Ther 2020, 5(1), 8.

2. Fig. 2B, for Hoechst stain, add small white arrows pointing to cells with condensed/fragmented nuclei. PI red cells are not necessarily necrotic; more accurate to say “dead” cells.

Response: Thank you for your suggestion. We have already revised it.

3. Evidence to suggest that DH32 or RT increases proteasomal degradation of beta-catenin resulting in decrease of stem cell transcription factors is weak. The result in Fig. 6c is not convincing of a big difference. Perhaps suggest as potential mechanism that needs further investigation.

Response: Thank you for your valuable suggestion. In our results of immunoprecipitation (IP) showed same intensity for heavy chain IgG. But the ubiquitin-β-catenin degradation showed significantly increased intensity in DH_32 treated NSCLCs (A549, H23 and H292) than the control cells and RT-treated NSCLCs cells (Figure 6C). 

4. Oct4 and Nanog forward primers used for qRT-PCR do not match in a Blast search. All other primer sequences are correct. Please clarify.

Response: Thank you for your suggestion. This OCT4 and NANOG forward primers sequences used for qRT-PCR in following research.

  1. Roy,S.; Lu,K.; Nayak,M.K.; Bhuniya,A.; Ghosh, T.; Kundu, S.; et al. Activation of D2 dopamine receptors in CD133+ve cancer stem cells in non-small cell lung carcinoma inhibits proliferation, clonogenic ability, and invasiveness of these cells. J Biol Chem 2017, 292(2), 435-445.
  2. Srinivasan, D.; Senbanjo, L.; Majumdar, S.; Franklin, R.B.; Chellaiah, M.A. Androgen receptor expression reduces stemness characteristics of prostate cancer cells (PC3) by repression of CD44 and SOX2. J Cell Biochem 2019, 120(2), 2413-2428.
  3. Page, R.L.; Ambady, S.; Holmes, W.F.; Vilner, L.; Kole, D.; Kashpur, O.; et al. Induction of stem cell gene expression in adult human fibroblasts without transgenes. Cloning Stem Cells 2009, 11(3), 417-426.
  4. Zhou, Y.; Kang, G.; Wen, Y.; Briggs, M.; Sebastiano, V.; Pederson, R.; et al. Do induced pluripotent stem cell characteristics correlate with efficient in vitro smooth muscle cell differentiation? A comparison of three patient-derived induced pluripotent stem cell lines. Stem Cells Dev 2018, 27(20), pp.1438-1448.
  5. Cardinale, V.; Renzi, A.; Carpino, G.; Torrice, A.; Bragazzi, M.C.; Giuliante, F.; et al. Profiles of cancer stem cell subpopulations in cholangiocarcinomas. Am J Pathol 2015, 185(6), 1724-1739.
  6. Luciani, M.; Garsia, C.; Beretta, S.; Petiti, L.; Peano, C.; Merelli, I.; et al. Human iPSC-derived neural stem cells display a radial glia-like signature in vitro and favorable long-term safety in transplanted mice. bioRxiv 2023, 2023-08.

5. Most important, there is no mention of potential use of RT or DH32 in animal models of lung cancer. It is critical to determine if RT or DH32 can be used in vivo and whether pharmacokinetics and bioavailability are favorable for use. Otherwise, the value of the work is diminished.

Response: Thank you for your suggestion. The finding of cancer stem cells (CSCs) targeted therapy is a new and pioneering strategy for the effective eradication of NSCLCs (1). Because the CSCs subpopulation of NSCLCs, possess remarkable capability in proliferation, self-renewal, and differentiation contribute to multiple tumor malignancies, such as recurrence, metastasis, heterogeneity, multidrug resistance, and radiation resistance. CSCs have generated substantial interest as a central focus for therapeutic intervention and treatment of resistant NSCLC cells (2-4).

Hence, our research elucidated the primarily targeted mechanism of action for the anti-cancer activity of compounds derived from right-half RT derivatives, DH_32. The mechanism of action revealed that in vitro human NSCLCs, DH_32, targeted to inhibit CSCs markers (CD133, CD44, ALDH1A1) and transcription factors (OCT4, NANOG, SOX2) by facilitating the degradation of β-catenin through ubiquitin-proteasomal degradation. This is the crucial stage in lead compound identification involves conducting in vitro experiments, and our emphasis is on discovering a novel lead compound with a mechanism of action that targets a specific protein.

When exploring pioneer compounds for CSC-targeted drugs, identifying a compound with potential designates it as a lead compound. This lead compound serves as the starting point, initiating subsequent optimization and chemical modifications for the development of a potential drug. The identification of a lead compound represents a critical and early step in the drug development process, laying the foundation for further refinement in the pursuit of effective cancer stem cell-targeted therapies (5).

For in vivo study is our upcoming strategy to advance the development of these compounds' right-half RT derivatives, DH_32. As the conventional protocol continues to recommend the injection or transplantation of treated cells into the animal model for assessing the efficacy of the compounds, the next step involves conducting experiments in an in vivo model. Following the confirmation of efficacy in the in vivo setting, additional investigation is conducted on pharmacokinetics and bioavailability. Your suggestion is very valuable in the concept of drug development, and this will be evaluated in further study.

Therefore, the potential targeted drugs for CSCs have been focused on evaluating new therapeutic systems for studying preclinical and clinical trials (6, 7). 

References

  1. Shibata, M.; Hoque, M.O. Targeting cancer stem cells: a strategy for effective eradication of cancer. Cancers 2019, 11(5), 732.
  2. Phi, L.T.H.; Sari, I.N.; Yang, Y.G.; Lee, S.H.; Jun, N.; Kim, K.S.; et al. Cancer stem cells (CSCs) in drug resistance and their therapeutic implications in cancer treatment. Stem Cells Int 2018, 2018.
  3. Zeng, Z.; Fu, M.; Hu, Y.; Wei, Y.; Wei, X.; Luo, M. Regulation and signaling pathways in cancer stem cells: implications for targeted therapy for cancer. Mol Cancer 2023, 22(1), 172.
  4. Cortes-Dericks, L.; Galetta, D. Impact of cancer stem cells and cancer stem cell-driven drug resiliency in lung tumor: options in sight. Cancers 2022, 14(2), 267.
  5. Hughes, J.P.; Rees, S.; Kalindjian, S.B.; Philpott, K.L. Principles of early drug discovery. Br J Pharmacol 2011, 162(6), 1239-1249.
  6. Yang, X.G.; Zhu, L.C.; Wang, Y.J.; Li, Y.Y.; Wang, D. Current advance of therapeutic agents in clinical trials potentially targeting tumor plasticity. Front Oncol 2019, 9, 887.
  7. Kumar, V.E.; Nambiar, R.; De Souza, C.; Nguyen, A.; Chien, J.; Lam, K.S. Targeting epigenetic modifiers of tumor plasticity and cancer stem cell behavior. Cells 2022, 11(9), 1403.

Comments on the Quality of English Language

Some English editing is required.

Response: Thank you for your suggestion. We have already revised it.

Reviewer 3 Report

Comments and Suggestions for Authors

The manuscript entitled "Simplified Synthesis of Renieramycin T Derivatives to Target Cancer Stem Cells via β-Catenin Proteasomal Degradation in Human Lung Cancer " by Ei et al. investigated the anti-cancer bioactivities of six analogs of Renieramycin T in lung cancer cell lines, and identified DH_32 as potent anti-tumor compound by inducing apoptosis, inhibiting proliferation and self-renewal. Overall, this is an interesting study that detailly studied the anti-tumor potential of the right half of Renieramycin T. However, there are some issues that need to be addressed to clarify the findings. Since my research area is in biology, I will only comment on the biological part.

Main points:

1.         How was the non-toxic concentration for DH_32 determined? In Figure 2A, how many doses of RT analogs were tested?  It is difficult to draw a conclusion from Figure 2A that 100 nM of DH_32 is non-toxic.

2.         If beta-catenin is believed to be the upstream regulator of the cancer stemness phenotypes, why did the authors choose to treat cells for 24 hours to examine beta-catenin in Figure 6, but OCT4 etc. for 6 hours in Figure 5? 

3.         Lack of a normal cell control in toxicity experiments. 

Minor:

1.         Lack of elongation temperature for real-time PCR. The primer Tm for NANOG-R is only 55.3 degrees, why did the authors use 60 degrees as annealing temperature? 

2.         Line 917, beta-actin serves as the loading control, “rabbit” is where the primary antibody was produced.

3.         Line 938, do you mean beta-catenin antibody?

4.         Line 943, the beads were resuspended in lysis buffer or sample buffer, or a certain ratio of both?

5.  Please provide better qualiity figures in Figure 4B.

Comments on the Quality of English Language

Overall, it is written in clear, readable, and understandable English.

Author Response

Reviewer 3

Comments and Suggestions for Authors

The manuscript entitled "Simplified Synthesis of Renieramycin T Derivatives to Target Cancer Stem Cells via β-Catenin Proteasomal Degradation in Human Lung Cancer " by Ei et al. investigated the anti-cancer bioactivities of six analogs of Renieramycin T in lung cancer cell lines, and identified DH_32 as potent anti-tumor compound by inducing apoptosis, inhibiting proliferation and self-renewal. Overall, this is an interesting study that detailly studied the anti-tumor potential of the right half of Renieramycin T. However, there are some issues that need to be addressed to clarify the findings. Since my research area is in biology, I will only comment on the biological part.

Main points:

  1. How was the non-toxic concentration for DH_32 determined? In Figure 2A, how many doses of RT analogs were tested? It is difficult to draw a conclusion from Figure 2A that 100 nM of DH_32 is non-toxic.

Response: Thank you for your suggestion. In Figure 2A, the IC50 of right-half RT analogs was determined using various doses (0, 0.5, 1, 10, 50, 100 μM).

We have same agreed with Reviewer 1 and Reviewer 1 suggested to change (‘non-toxic concentration) to (‘low concentration).

  1. If beta-catenin is believed to be the upstream regulator of the cancer stemness phenotypes, why did the authors choose to treat cells for 24 hours to examine beta-catenin in Figure 6, but OCT4 etc. for 6 hours in Figure 5?

Response: Thank you for your suggestion. Figure 5A depicts the assessment of mRNA levels for stem cell transcription factors (OCT4, NANOG, SOX2) in A549, H23, and H292 cells after a 6 h treatment with DH_32, as measured using RT-qPCR method. The Western blot analysis conducted at 24 h assesses the protein expression level of the upstream regulator (beta-catenin) linked to cancer stemness phenotypes. According to existing literature, the OCT4, etc. mRNA level measure at 6 h by RT-qPCR method (1).   

Reference

  1. Chen, M.; Ye, A.; Wei, J.; Wang, R.; Poon, K. Deoxycholic acid upregulates the reprogramming factors KFL4 and OCT4 through the IL-6/STAT3 pathway in esophageal adenocarcinoma cells. Technol. Cancer Res Treat 2020, 19, 1533033820945302.
  2. Lack of a normal cell control in toxicity experiments.

Response: Thank you for your suggestion. We already added IC50 value for non-tumorigenic epithelial cell line from human bronchial epithelium cells (BEAS2B) and the human dermal papilla cells (DP) were treated with DH_32 in Figure 2C.

Minor:

1. Lack of elongation temperature for real-time PCR. The primer Tm for NANOG-R is only 55.3 degrees, why did the authors use 60 degrees as annealing temperature?

Response: Thank you for your suggestion. We have already revised the RT-qPCR method to address the elongation temperature.

According to our results, the Ct cycle for NANOG is about 25. This Ct cycle values indicated that strong positive reactions indicate abundant target nucleic acid in sample (1). Kojima et al showed RT-qPCR condition for NANOG gene (2).

References

  1. https://www.wvdl.wisc.edu/wpcontent/uploads/2013/01/WVDL.Info_.PCR_Ct_Values1.pdf
  2. Kojima, H.; Okumura, T.; Yamaguchi, T.; Miwa, T.; Shimada, Y.; Nagata, T. Enhanced cancer stem cell properties of a mitotically quiescent subpopulation of p75NTR-positive cells in esophageal squamous cell carcinoma. Int J Oncol 201751(1), 49-62.

2. Line 917, beta-actin serves as the loading control, “rabbit” is where the primary antibody was produced.

Response: Thank you for your suggestion. Rabbit beta-actin (cat no: 4970) was obtained from cell signaling (Beverly, MA, USA).

3. Line 938, do you mean beta-catenin antibody?

Response: Thank you for your suggestion. Yes, this means beta-catenin antibody.

4. Line 943, the beads were resuspended in lysis buffer or sample buffer, or a certain ratio of both?

Response: Thank you for your suggestion. The magnetic bead-antibody-antigen complex beads was re-suspended in lysis buffer (30 μL), and the 6x sample buffer (5 μL) was denatured by heating at 95˚C for 5 min.

5. Please provide better qualiity figures in Figure 4B.

Response: Thank you for your suggestion. We have already revised the quality for Figure 4B.

Comments on the Quality of English Language

Overall, it is written in clear, readable, and understandable English.

Response: Thank you for your suggestion.

Round 2

Reviewer 3 Report

Comments and Suggestions for Authors

The revised manuscript entitled "Simplified Synthesis of Renieramycin T Derivatives to Target 

Cancer Stem Cells via β-Catenin Proteasomal Degradation in Human Lung Cancer " by Ei et al. addressed most of my concerns except main point 2 and minor point 1. 

 My original main point 2 is:

If beta-catenin is believed to be the upstream regulator of the cancer stemness phenotypes, why did the authors choose to treat cells for 24 hours to examine beta-catenin in Figure 6, but OCT4 etc. for 6 hours in Figure 5? 

Response: Thank you for your suggestion. Figure 5A depicts the assessment of mRNA levels for stem cell transcription factors (OCT4, NANOG, SOX2) in A549, H23, and H292 cells after a 6 h treatment with DH_32, as measured using RT-qPCR method. The Western blot analysis conducted at 24 h assesses the protein expression level of the upstream regulator (beta-catenin) linked to cancer stemness phenotypes. According to existing literature, the OCT4, etc. mRNA level measure at 6 h by RT-qPCR method (1).   

Reference

  1. Chen, M.; Ye, A.; Wei, J.; Wang, R.; Poon, K. Deoxycholic acid upregulates the reprogramming factors KFL4 and OCT4 through the IL-6/STAT3 pathway in esophageal adenocarcinoma cells. Technol. Cancer Res Treat 2020, 19, 1533033820945302.

My point is that the signaling change (measured at the protein level) should occur earlier or at least at the same time as the downstream functional events (measured by downstream transcriptional activity by real-time PCR). So the authors should measure the signaling events before or at 6 hours post treatment.

My original minor point 1 is:

Lack of elongation temperature for real-time PCR. The primer Tm for NANOG-R is only 55.3 degrees, why did the authors use 60 degrees as annealing temperature? 

Response: Thank you for your suggestion. We have already revised the RT-qPCR method to address the elongation temperature.

According to our results, the Ct cycle for NANOG is about 25. This Ct cycle values indicated that strong positive reactions indicate abundant target nucleic acid in sample (1). Kojima et al showed RT-qPCR condition for NANOG gene (2).

References

  1. https://www.wvdl.wisc.edu/wpcontent/uploads/2013/01/WVDL.Info_.PCR_Ct_Values1.pdf
  2. Kojima, H.; Okumura, T.; Yamaguchi, T.; Miwa, T.; Shimada, Y.; Nagata, T. Enhanced cancer stem cell properties of a mitotically quiescent subpopulation of p75NTR-positive cells in esophageal squamous cell carcinoma. Int J Oncol 201751(1), 49-62.

The new text in the revised manuscript is: 

For the RT-qPCR, the protocol included an initial denaturation step at 95 ̊C for 1 min, followed by 45 cycles involving denaturation at 95 ̊C for 10 sec and primer extension at 60 ̊C for 30 sec. 

My original point is that the authors only gave the annealing temperature but not elongation one. In the revised text, they delete the annealing and change it to elongation, which is not acceptable.

The PCR cycle has three steps: denaturation, annealing, extension. In the most standard protocol in RT-qPCR reaction, the annealing and extension temperature are the same, which is 60 ̊C. This is OK, the authors should just say that annealing and extension are at 60 ̊C.

The problem is the Tm for primers, especially for the NANOG reverse primer.  The Tm is the temperature at which the primer can anneal to the template. Above Tm, the primer will usually not be able to bind stably to the template and therefore will not give enough amplification. It is possible that the RT-qPCR machine and the primer synthesis company used different methods to calculate the Tm, but it won’t make that much difference. I suggest the authors try another way to calculate the Tm for primers, because the Tm they now show is misleading to readers. Related question is primers for GAPDH, they are near 70 ̊ C, which means at 60 ̊C annealing temperature, the unspecific binding can happen with greater possibility. It is best to have the primer sets with Tm as close as possible.

Author Response

Reviewer 3

The revised manuscript entitled "Simplified Synthesis of Renieramycin T Derivatives to Target Cancer Stem Cells via β-Catenin Proteasomal Degradation in Human Lung Cancer " by Ei et al. addressed most of my concerns except main point 2 and minor point 1.

My original main point 2 is:

If beta-catenin is believed to be the upstream regulator of the cancer stemness phenotypes, why did the authors choose to treat cells for 24 hours to examine beta-catenin in Figure 6, but OCT4 etc. for 6 hours in Figure 5?

Response: Thank you for your suggestion. Figure 5A depicts the assessment of mRNA levels for stem cell transcription factors (OCT4, NANOG, SOX2) in A549, H23, and H292 cells after a 6 h treatment with DH_32, as measured using RT-qPCR method. The Western blot analysis conducted at 24 h assesses the protein expression level of the upstream regulator (beta-catenin) linked to cancer stemness phenotypes. According to existing literature, the OCT4, etc. mRNA level measure at 6 h by RT-qPCR method (1).   Reference

Chen, M.; Ye, A.; Wei, J.; Wang, R.; Poon, K. Deoxycholic acid upregulates the reprogramming factors KFL4 and OCT4 through the IL-6/STAT3 pathway in esophageal adenocarcinoma cells. Technol. Cancer Res Treat 2020, 19, 1533033820945302.

My point is that the signaling change (measured at the protein level) should occur earlier or at least at the same time as the downstream functional events (measured by downstream transcriptional activity by real-time PCR). So the authors should measure the signaling events before or at 6 hours post treatment.

New Response: Thank you for your valuable suggestion. In our experiments, the mRNA expression of OCT4, etc. was detected at 6 h after beta-catenin expression increased. Therefore, our total treatment time for DH_32 with NSCLCs is (24 h + 6 h) = 30 h.

 My original minor point 1 is:

Lack of elongation temperature for real-time PCR. The primer Tm for NANOG-R is only 55.3 degrees, why did the authors use 60 degrees as annealing temperature?

Response: Thank you for your suggestion. We have already revised the RT-qPCR method to address the elongation temperature. According to our results, the Ct cycle for NANOG is about 25. This Ct cycle values indicated that strong positive reactions indicate abundant target nucleic acid in sample (1). Kojima et al showed RT-qPCR condition for NANOG gene (2).

References

https://www.wvdl.wisc.edu/wpcontent/uploads/2013/01/WVDL.Info_.PCR_Ct_Values1.pdf

Kojima, H.; Okumura, T.; Yamaguchi, T.; Miwa, T.; Shimada, Y.; Nagata, T. Enhanced cancer stem cell properties of a mitotically quiescent subpopulation of p75NTR-positive cells in esophageal squamous cell carcinoma. Int J Oncol 2017, 51(1), 49-62.

The new text in the revised manuscript is:  For the RT-qPCR, the protocol included an initial denaturation step at 95 ̊C for 1 min, followed by 45 cycles involving denaturation at 95 ̊C for 10 sec and primer extension at 60 ̊C for 30 sec.

My original point is that the authors only gave the annealing temperature but not elongation one. In the revised text, they delete the annealing and change it to elongation, which is not acceptable.

The PCR cycle has three steps: denaturation, annealing, extension. In the most standard protocol in RT-qPCR reaction, the annealing and extension temperature are the same, which is 60 ̊C. This is OK, the authors should just say that annealing and extension are at 60 ̊C.

The problem is the Tm for primers, especially for the NANOG reverse primer.  The Tm is the temperature at which the primer can anneal to the template. Above Tm, the primer will usually not be able to bind stably to the template and therefore will not give enough amplification. It is possible that the RT-qPCR machine and the primer synthesis company used different methods to calculate the Tm, but it won’t make that much difference. I suggest the authors try another way to calculate the Tm for primers, because the Tm they now show is misleading to readers. Related question is primers for GAPDH, they are near 70 ̊ C, which means at 60 ̊C annealing temperature, the unspecific binding can happen with greater possibility. It is best to have the primer sets with Tm as close as possible.

New Response: Thank you for your valuable suggestion. We have already revised the RT-qPCR method.

We have calculated the Tm for NANOG reverse primer is 58 ̊C.

We followed the RT-qPCR protocol according to the manufacturer's instructions for the Luna® Universal qPCR Master Mix Kit Protocol (M3003L).
